psychology

maths attainment, school transition, maths attitudes, school affect, student–teacher relationships, Avon Longitudinal Study of Parents and Children

**Author for correspondence:**
Danielle Evans
e-mail: de84@sussex.ac.uk

# Maths attitudes, school affect and teacher characteristics as predictors of maths attainment trajectories in primary and secondary education

## Danielle Evans and Andy P. Field

School of Psychology, University of Sussex, Brighton, UK

(iD) DE, 0000-0002-5330-3393; APF, 0000-0003-3306-4695

Maths attainment is essential for a wide range of outcomes relating to further education, careers, health and the wider economy. Research suggests a significant proportion of adults and adolescents are underachieving in maths within the UK, making this a key area for research. This study investigates the role of children's perceptions of the school climate (children's affect towards school and student–teacher relationships), their attitudes towards maths and teacher characteristics as predictors of maths attainment trajectories, taking the transition from primary to secondary education into consideration. Two growth models were fit using secondary data analysis of the Avon Longitudinal Study of Parents and Children (ALSPAC). The first model, which looked at predictors of maths attainment in primary education, found significant associations only between positive maths attitudes and increased maths attainment. The second model, which looked at predictors of maths attainment in secondary education, found significant associations between increased maths attainment and positive maths attitudes, decreased school belonging, positive student–teacher relationships and increased teacher fairness. The findings suggest that the secondary education school environment is particularly important for maths attainment.

# 1. Introduction

Aspects of numerical and mathematical skills are used by adults every day. Whether this is as employees giving the correct change or when using spreadsheets, as consumers when

calculating the savings associated with a 10% discount, when managing finances (i.e. understanding interest rates and borrowing funds), or as parents when helping children with homework [1]. The consequences of poor numeracy and low maths attainment are far-reaching and long-lasting. Low maths attainment limits educational and career opportunities, and is linked to a higher rate of unemployment and low socioeconomic status (SES), as well as increased health issues, and a higher likelihood of homelessness and contact with the criminal justice system [2–6]. Poor numeracy is reported to cost around 20.2 billion per year to the UK economy alone, not including the potential costs associated with the health sector and criminal justice system [7].

When quantifying the extent of poor mathematical abilities in the UK, it is reported that 49% of working-age adults have the equivalent maths skills of 6-year-old children, with only 22% having the skills of the 'average' 16-year-old [8]. However, these statistics are somewhat dated (using data from 2011), meaning that the true extent of the 'maths crisis' [9] currently is unknown. When comparing the data from 2011 with the first wave in 2003, the percentage of 'numerate' adults in the UK had decreased [8], suggesting that it would not be entirely illogical to assume that the problem has continued to worsen from 2011 until now. The poor maths performance seen in adults in the UK is likely  to be due to deficits in childhood learning but could also be due to poor retention or a lack of practice of maths skills over time (see [10]). Investigating predictors of maths attainment in childhood provides several benefits in helping to overcome the 'maths crisis' present in the UK. By uncovering underlying factors that influence maths attainment, we can use this information to design evidence-based strategies, which will hopefully increase the effectiveness of interventions aiming to improve maths attainment and other positive outcomes associated with increased abilities.

Estimates of the heritability of maths suggest that attainment is moderately genetic—around two-thirds of the variance in attainment can be explained by genetic factors, with the remaining variance explained by aspects of the shared- and non-shared environment [11], and their interaction with genetic factors. Outside of the home, a significant proportion of children's time is spent in school. It is within this environment that children acquire new knowledge and skills, and is also where significant social interactions with others take place. Unsurprisingly, existing research suggests the school environment is influential in the development of maths abilities and the performance of maths skills. However, the long-term effects are unknown. Therefore, the present study aims to investigate which school-related factors are longitudinally associated with maths attainment trajectories of school children in the UK, with a particular focus on the school climate, student–teacher relationships and maths-related attitudes during the transitional period from primary to secondary education.

## 1.1. The transition from primary to secondary education

Early adolescence is a period of substantial change and development. One key event associated with considerable disruption during this time is the transition from primary to secondary education. In the UK, this transition occurs when children are 11 years old when they transfer from their 6th year of education in a primary school to their 7th year in a separate secondary school. The transition event itself is negatively associated with academic, social and emotional wellbeing [12,13], with children experiencing increased feelings of anxiety, loneliness and stress during the transitional period [14–17]. Many changes occur within children's environments stemming from the transition, particularly relating to differences between primary and secondary education institutions. The differences present between these environments could plausibly influence relationships between maths attainment and the school climate, student–teacher relationships and attitudes towards maths around the transition to secondary education, which will be discussed further in the following sections.

One clear environmental difference is that secondary schools are typically much larger than primary schools, with several primary education institutions 'feeding' into one secondary school. Children generally have several specialized subject teachers in secondary education compared with one individual teacher for all subjects for the entire school year in primary education (though the presence of specialist maths teachers is becoming increasingly common in English primary schools, helped by government initiatives and training bursaries). Children report several concerns relating to this new environment, such as becoming lost when navigating their new school buildings or being late for class [18,19]. Prospective relationships in secondary education also cause some concern among children during the transition process, especially regarding bullying and making new student–teacher relationships [19]. The increased size of the physical and social environment, and the additional interactions between children and their unfamiliar teachers and peers probably affect children's

perceptions of the school climate and student–teacher relationships, and how these factors are associated with attainment.

This transitional period in adolescence is particularly interesting when investigating maths attainment trajectories because of the impact of the education transition and the differences found in maths outcomes between primary and secondary education students. For example, students in secondary education report less involvement in maths class, less positive attitudes towards maths, decreased maths enjoyment, decreased maths interest and increased maths anxiety compared with primary education students [20,21]. These 'attitudes' towards maths (i.e. interest, enjoyment, self-efficacy and anxiety) are linked to maths attainment [22–28], highlighting the importance of this period for intervention strategies aiming to improve maths attainment. Further research also demonstrates poor maths performance across the transition where declines in achievement and a lack of progress in maths has been found [29–31], which has been linked to increased maths anxiety at age 18 [32].

The changes found in attitudes towards maths (i.e. declining efficacy and interest) across the primary–secondary education transition appear to be related to aspects of the school environment, such as post-transition teacher effectiveness [20]. Midgley *et al.* [33] found that maths attitudes (i.e. value, usefulness and importance) significantly declined for students moving from teachers they perceived to be highly supportive pre-transition to teachers they perceived to be less supportive post-transition, which was particularly marked for low-achieving students. These findings together suggest that the wider school environment is especially important for maths-related outcomes across the transition to secondary education, and that the differing characteristics of primary and secondary education environments should be investigated further when assessing maths attainment in adolescence.

## 1.2. School-related predictors of maths

### 1.2.1. The school climate and children's affect towards school

The 'school climate' has been defined as the 'norms, goals, values, interpersonal relationships, teaching and learning practices, and organizational structures' of a school, and relatedly, children's affect towards school encompassing their feelings of social, emotional and physical safety [34, p. 182]. A positive school/classroom climate, favourable affect towards school and an academically focused environment is positively associated with children's general academic and maths attainment [35–40]. Students perceiving their classroom to be highly emotionally supportive are more likely to seek help from their teachers and peers, which consequently is related to increased maths attainment [41]. The school climate is also associated with adolescents' wellbeing [38,42], with increased feelings of 'school connectedness' associated with decreased emotional distress, suicidal involvement, violence and substance use in US adolescents [43].

There are several changes within the school environment that occur with the transition to secondary education which makes this transitional period particularly interesting in terms of the school climate. Children transition from being the oldest in the school to the youngest in a larger, very unfamiliar environment, probably affecting their sense of security. The total number of students also increases significantly from primary to secondary education, with teachers typically interacting with multiple classes of children in different years throughout the school day, meaning that children have a decreased capacity to develop close relationships and attachments like they had with their teachers in primary education [44]. These differences in the primary and secondary school environment, and those highlighted previously, could affect adolescents' perceptions of the school climate and their feelings towards school. It is likely that the change from a small classroom where children hold close relationships with their teachers, to a larger departmentalised school with an increased focus on discipline, affects their feelings of social, emotional and physical safety. Findings reported by Coelho *et al.* [45] support this idea, highlighting the negative impact of the primary–secondary education transition on school climate, with declines in ratings of peer relationships, fairness of rules, school safety, school liking and student–teacher relationships post-transition. However, the transition for Portuguese students in the study conducted by Coelho *et al.* is one of the earliest primary–secondary education transitions to occur at age 9 compared with age 11 in the UK, meaning the effects could potentially be different for older students. Although, in a study of US schools, Kim *et al.* [46] report that 'K-8' schools (i.e. schools that do not transition in grade 6 or 7) had a more positive social context (characterized by school chaos, student conduct problems, staff professional climate, teacher agency and teaching burden) compared with middle and junior high schools that do transition (usually into grade 6 or 7), suggesting that the negative impacts associated with the transition to secondary

education reported by Coelho *et al.* are also evident in older adolescents. Positive school affect also appears to have a protective role; Vaz *et al.* [47] report an increased sense of 'school-belonging' (an aspect of school climate) in primary school is associated with decreased emotional symptoms concurrently, and in the first year of secondary education.

### 1.2.2. Student–teacher relationships

Overall, the existing literature suggests that a positive school climate is important for children's academic attainment and their socio-emotional functioning. The aforementioned studies have used a range of definitions for 'school climate'; however, one aspect that is commonly investigated within the school-climate literature is the relationship students have with their teachers. Various aspects of student–teacher relationships have been examined, though most studies focus on closeness, warmth, trust and fairness perceived by students. Previous research has found that positive and warm student–teacher relationships buffer the effects of childhood adversity on aspects of cognitive abilities [48], and protect against depressive symptoms and misconduct in adolescents [49]. Positive student–teacher relationships are associated with lower dropout rates for US high-school students [50] and influences student wellbeing [51]. Others report associations between teacher mental health problems and students' mental wellbeing [52].

As well as being important for general student wellbeing, positive student–teacher relationships also play a pivotal role within maths attainment [53]. Increased student–teacher 'connectedness' is associated with increased maths attainment in Canadian adolescents, and also has a buffering effect between bullying and maths attainment for boys [54]. Positive student–teacher relationships have been found to mediate the effects of school-level poverty on maths achievement in Chinese students [55]. Teng [56] supports this finding, highlighting the importance of student–teacher relationships for the maths attainment of Chinese adolescents, with a marked effect for low-performing schools and underachievers. Negative relationships also appear to have an effect on attainment. Bryce *et al.* [57] found student–teacher conflict negatively impacted academic achievement (maths and reading) through behavioural engagement in US school children.

In addition to the effects associated with a positive student–teacher relationship, teachers' own attitudes, self-efficacy beliefs and abilities can influence the development of students' attitudes towards maths regarding gender stereotypes [58], and can affect students' attainment [58,59]. Teachers' enjoyment of maths also affects the instructional time given to maths in that teachers who enjoy maths more spend more time engaging in maths tasks [60]. There is also some evidence to suggest teachers' general mental wellbeing is linked to students' maths abilities through the quality of the classroom learning environment [61], and in the feedback given to students [62], which is particularly marked for low-achieving students. Research in this area is sparse, but generally suggests that teachers' mental health and their attitudes towards maths are linked to students' maths outcomes.

Similar to the school climate, changes in student–teacher relationships have been reported around the transition to secondary education. In primary education, children are traditionally taught by a single teacher per year for all subjects (though primary schools are increasingly using specialist teachers in recent years) whereas, in secondary education, the majority of institutions are departmentalized in that adolescents will be taught different subjects by different specialist teachers. This difference between primary and secondary education is thought to alter the relationships students and teachers have [63]. For example, Hughes & Cao [64] report a significant drop in teacher-rated 'warmth' around the transition to secondary education for US adolescents, with larger decreases in warmth predictive of lower maths attainment. Alternatively, Bru *et al.* [65] report no abrupt change in student-perceived teacher support around the transition. These differences in findings could potentially lie within the respondent (student or teacher), the sample used (US versus Norway) or the specific aspects of the student–teacher relationship investigated, further highlighting the complexity of this association, and the need for further research in this area.

## 1.3. Stage–environment fit theory

One theoretical framework that may help to explain the negative effects and outcomes associated with the transition into secondary education is the stage–environment fit theory proposed by Eccles *et al.* [44]. This theory states that negative outcomes occur when there is a mismatch between adolescents' needs and the opportunities within their environments. Eccles *et al.* propose that there are developmentally inappropriate changes within the school and classroom environment following the transition to secondary education, which may result in a poor person–environment fit. The changes discussed by

Eccles *et al.* [44] include a greater emphasis on teacher control and discipline, decreased opportunities for decision-making and responsibilities in class, fewer positive student–teacher relationships, whole-group task instruction (i.e. all students completing the same tasks in class and for homework assignments which increases social comparison, competitiveness and evaluation concerns), public forms of evaluation and normative grading systems, and decreasing cognitive demands (i.e. through work involving copying from the board or textbooks). These changes are proposed to be damaging to motivational constructs post-transition, and can therefore potentially affect attainment and socio-emotional adaptation to secondary education which could have long-lasting implications. Based on these changes, and the poor fit between adolescents' needs and those provided by the secondary education environment, it is plausible that the relationships between school-related predictors and maths attainment will differ between primary and secondary education.

## 1.4. The present study

To summarize, the transition to secondary education is regarded as a particularly stressful period for young adolescents. The transition coincides with biological, psychological, environmental and social changes and is associated with negative outcomes, especially where adolescents fail to adapt to their new environment successfully. Various aspects of the school environment are associated with maths attainment, including the school climate, student–teacher relationships and children's attitudes towards maths (often associated with teacher attitudes). These aspects are thought to differ substantially between primary and secondary education, often reported as a consequence of the transition event. However, there is an absence of research exploring the effects of the school climate (children's affect towards school and teacher characteristics), student–teacher relationships and attitudes towards maths on maths attainment in primary and secondary education with little known of the potential long-term effects. Given the alarming state of the maths abilities of children and adults in the UK currently, identifying predictors of attainment early in development is important to allow for effective interventions. Therefore, the present study aims to explore the aforementioned factors as predictors of maths attainment trajectories in primary and secondary education.

This study presents two growth models examining variables in primary and secondary education. The models use secondary data from the Avon Longitudinal Study of Parents and Children (ALSPAC) to investigate predictors of the maths attainment trajectories (from age 7 to 16) of UK students. ALSPAC has been used in previous studies investigating school-related factors including risk factors for school exclusion [66], the effects of peer victimization [67] and examining school-related protective factors against negative outcomes for children experiencing maltreatment in early childhood [68]. The current study is the final part of a three-phase study looking at predictors of maths attainment using the ALSPAC sample. The previous two phases [69,70], which focused on the home environment, parental, cognitive and emotional factors, showed that working memory, internalizing symptoms, parent–child relationships, parental education and school involvement significantly predict maths attainment. The current final phase focuses on school-related predictors of maths attainment trajectories. The primary education model investigates the effects of children's affect towards school, relationships with teachers, attitudes towards maths and primary education teacher characteristics (affect towards teaching, mental wellbeing and self-esteem). The secondary education model investigates the effects of school belonging, negative emotion towards school, relationships with teachers, attitudes towards maths and children's feelings towards their secondary education maths teacher. Primary education variables and secondary education variables are analysed separately as they are not comparable across the transition. It is hypothesized that greater positive affect towards the school environment, positive student–teacher relationships and favourable attitudes towards maths and maths teachers will be associated with increased attainment in both primary and secondary education. It is predicted that teachers' self-rated characteristics (affect towards teaching, mental wellbeing and self-esteem) will predict maths attainment where increased self-esteem and fewer mental health symptoms will be associated with increased attainment.

# 2. Method

## 2.1. Sample

The Avon Longitudinal Study of Parents and Children (ALSPAC) recruited expectant mothers residing in the South West of England, due to give birth between 1 April 1991 and 31 December 1992 [71,72]. The

core ALSPAC sample consisted of 14 062 live births, of which 13 988 children were alive at 1 year. ALSPAC also recruited additional participants post-birth which resulted in a total sample size of 15 589 fetuses, of which 14 901 children were alive at 1 year. The sample is generally representative; however, there is a slight over-representation of white families with higher SES [71].

Data were collected from the child, the child's mother and her partner, and the child's school teachers, as well as education-linked data from the National Pupil Database (NPD). The majority of the data were collected through self-report postal questionnaires, with some of the data collected in 'Children in Focus' clinics, which a smaller subsample (10%) were invited to attend.

The study website contains details of all the data that are available through a fully searchable data dictionary and variable search tool (see http://www.bristol.ac.uk/alspac/researchers/our-data/). All participants provided written informed consent prior to the study. Ethical approval was obtained from the ALSPAC Ethics and Law Committee and the Local Research Ethics Committees. Informed consent for the use of data collected via questionnaires and clinics was obtained from participants following the recommendations of the ALSPAC Ethics and Law Committee at the time.

### 2.1.1. Sample exclusions

The sample exclusions here are the same as those in the first two phases of the project, presented in Evans *et al.* [70] and Evans & Field [69]. Only data for singletons and the first-born twin were retained for analysis. Children identified as having special educational needs (SEN) at age 7 and/or age 11 were also excluded from analysis, as were children with English as an additional language (combined $n = 2666$). Attrition was particularly high due to the longitudinal design, so participants lacking sufficient data (i.e. those with at least 50% missing data for the predictor variables) were excluded from analysis, leading to a final sample size of 6490.

## 2.2. Outcome

### 2.2.1. Maths attainment

There are four key stages throughout children's compulsory education in England, with key stage 1 (age 5–7) and key stage 2 (age 7–11) in primary education, and key stage 3 (age 11–14) and key stage 4 (age 14–16) in secondary education. The maths attainment of primary and secondary education students is measured through examinations and assessments at the end of each key stage (i.e. at age 6–7, 10–11, 13–14 and 15–16).

In key stages 1–3, children's progress is evaluated using national curriculum levels which are numerical grades ranging from 1 to 8, with a higher score indicative of greater maths attainment. Governmental guidelines suggest that it is expected that children achieve a level 2 at key stage 1, a level 4 at key stage 2 and between levels 5 and 6 at key stage 3. At key stage 4, adolescents can achieve an alphabetical grade from the highest of 'A*', through 'A', 'B', 'C', 'D', 'E', 'F', 'G' and the lowest grade of a 'U'. To be comparable with maths attainment at the other key stages, these alphabetical grades were coded into numerical grades with the highest being grade 10 (i.e. 'A*'), down to the lowest grade of 2 (i.e. 'U'). National curriculum levels for maths were obtained by ALSPAC from local education authorities for key stage 1 data, and the NPD for key stage 2–4 data (NPD variables: K2_LEVM, K3_LEVM and KS4_APMAT), which consisted of a combination of teacher assessments and standardized tasks and tests. It is important to highlight that this scoring differs from the current grading system in England where key stage 3 tests are no longer administered, and where key stage 4 assessments are graded on a 1–9 scale.

In this study, the main outcomes were maths attainment in primary education just prior to the transition to secondary education (age 11; key stage 2), maths attainment post-transition to secondary education (age 14; key stage 3) and the growth in maths attainment over time.

## 2.3. Substantial predictors: primary education

Where measures were not pre-existing, validated questionnaires, measures were constructed from items in the ALSPAC dataset relating to common constructs. In these cases, a polychor factor analysis and parallel analyses were used to determine items that could be combined. The *polychor* [73] and *nFactors* [74] packages in R were used for these analyses and the *psych* [75] package was used to determine

internal consistency. A table of all the individual items for each of the measures where composites were created is available in the electronic supplementary material.

### 2.3.1. Primary school affect

Children's feelings towards primary school were assessed at age 11. Children were asked to report their feelings towards school and teachers by stating their agreement with 11 statements on a 4-point scale (*disagree, somewhat disagree, somewhat agree* and *agree*; scored as 0–3). Example statements included: 'my school is a place where my teacher listens to what I say', 'my school is a place where other pupils are very friendly' and 'my school is a place where I feel worried'.

A polychoric factor analysis revealed two factors determined by parallel analysis, relating to affect towards school, and relationships with teachers. Composites were created summing the scores for the items making up each factor, with possible scores for affect towards school ranging from 0 to 24 (8 items; such as 'my school is a place where I get on well with the other pupils in my class'), and possible scores for relationships with teachers ranging from 0 to 9 (3 items; such as 'my school is a place where my teacher treats me fairly in class'). A higher score indicates more positive affect towards school and teachers for both measures. Reliability was moderately high; Cronbach's $\alpha$ was 0.80 and 0.75 for affect towards school and relationships with teachers, respectively.

### 2.3.2. Attitudes to maths (age 10)

Children's attitudes towards maths in primary education were measured at age 10. Children were asked to rate their enjoyment, interest and abilities in maths by responding to 10 items on a 5-point scale (*not true, somewhat untrue, partly true, somewhat true* and *true*; scored as 0–4). Example items include: 'I get good marks in maths', 'I enjoy doing work in maths' and 'I am bad at maths'. The responses were coded in a way that a higher score indicated more positive attitudes towards maths. Polychoric factor analysis and parallel analysis revealed a single factor; therefore, a composite was created summing the responses to all 10 items with possible scores ranging from 0 to 40. Cronbach's $\alpha$ was high at 0.95.

### 2.3.3. Primary education teacher characteristics

Measures related to teacher characteristics were assessed in the final year of primary education (in year 6; when children are age 10–11). Three variables were included, consisting of the teacher's feelings towards teaching (teacher affect), their mental health and their self-esteem. The teacher's affect towards teaching was measured by asking teachers to state their agreement (on a 5-point scale from *strongly disagree* to *strongly agree*; scored as 0–4) with six statements broadly covering their enjoyment of teaching, their confidence in and enjoyment of teaching numeracy, and how much they find teaching worthwhile. Polychoric factor analysis and parallel analysis revealed one factor for teacher affect, meaning a composite could be made. The score for teacher's affect was made from summing the scores for the six items, with a higher score referring to more positive affect towards teaching (ranging from 0 to 24). Cronbach's $\alpha$ was adequate at 0.71.

Teacher mental health and self-esteem were measured using the Crown-Crisp Experiential Index (CCEI) and the Bachman Self Esteem score. The CCEI [76] used by ALSPAC contains 23 items relating to somatic, depressive and anxious symptoms. Possible scores range between 0 and 46, with a higher score corresponding to more symptoms. The Bachman Self Esteem score [77] consists of 10 questions with a possible score between 0 and 40. A higher score relates to higher self-esteem. Cronbach's $\alpha$ for the aforementioned CCEI subscales ranges from 0.66 to 0.79 [78], and $\alpha$ for the Bachman Self Esteem score is 0.75 [77].

## 2.4. Substantial predictors: secondary education

### 2.4.1. Secondary school affect

Feelings towards secondary school were measured at age 14. Adolescents were given the same 11 statements as the primary school affect measure above, and were asked to rate their agreement with the statements on a 4-point scale (*strongly disagree, disagree, agree* and *strongly agree*; scored as 0–3). Example statements included: 'my school is a place where I get on well with other pupils in my classes', 'my school is a place where I feel proud to be a pupil' and 'my school is a place where I feel lonely'. A polychoric factor analysis was conducted on the 11 items and parallel analysis revealed

three factors relating to school belonging, negative emotion towards school and relationships with teachers. Composites were created summing the scores for the items making up each of the three factors. Possible scores for school belonging (6 items) ranged from 0 to 18, for negative emotion (3 items) scores ranged between 0 and 9 and for relationships with teachers (2 items) scores ranged from 0 to 6. A higher score indicates greater school belonging, less negative emotion towards school and more positive relationships with teachers. School belonging had high reliability (Cronbach's $\alpha = 0.83$), relationships with teachers had moderately high reliability (Cronbach's $\alpha = 0.72$), and negative emotion towards school had adequate reliability (Cronbach's $\alpha = 0.64$).

### 2.4.2. Attitudes to maths (age 14)

Maths attitudes in secondary education were assessed at age 14. Adolescents were asked to indicate how much they enjoyed doing maths, how much they find what they learn in maths useful and the level of importance they place on being good at maths. Maths enjoyment and usefulness were measured on 5-point scales (i.e. *doesn't like it at all* to *likes it very much* and *not very useful* to *very useful*; scored as 0–4). The level of importance adolescents placed on being good at maths was measured on a 4-point scale from *not at all important* to *very important* (scored as 0–3), which was recoded to be on a 5-point scale without a neutral option. A polychoric factor analysis was conducted on the three items and revealed one factor (using parallel analysis); therefore, a composite was made. The scores for each of the items were summed together with possible scores ranging from 0 to 12, with a higher score indicating positive attitudes towards maths. Reliability was moderately high; Cronbach's $\alpha = 0.68$.

### 2.4.3. Feelings towards maths teacher

At age 14, adolescents were given 18 statements regarding their feelings and perceptions of their maths teacher, and were asked to rate their feelings on a 5-point scale (from *strongly disagree* to *strongly agree*; scored as 0–4). Using polychoric factor analysis and parallel analysis, two factors were found from the 18 items relating to positive teaching (12 items), and teacher fairness towards pupils (5 items). The scores for the individual items were summed to create two factors, for positive teaching scores could range from 0 to 48 and for teacher fairness possible scores could range from 0 to 20. Positive teaching included statements of teacher competence and measures of positive teaching practices such as 'my maths teacher understands maths really well', 'everyone is encouraged to do their very best' and 'my maths teacher gives us time to really explore and understand new things'. For teacher fairness, example items included: 'my maths teacher only cares about the clever students', 'my maths teacher treats boys and girls differently' and 'my maths teacher treats some students better than other students'. A higher score on both measures indicates more positive teaching practices and greater perceived teacher fairness towards pupils. High reliability was found for both measures, Cronbach's $\alpha = 0.90$ and 0.85 for positive teaching practices and teacher fairness, respectively.

## 2.5. Contextual predictors

### 2.5.1. Biological sex

Biological sex was recorded at birth, and included as a predictor due to potential differences in maths attainment between males and females. Females accounted for 55.3% of the sample. In both models, females were used as the reference group.

### 2.5.2. Socioeconomic status

During the mother's pregnancy (at 32 weeks gestation), SES of both parents (where available) was assessed using the Cambridge Social Interaction and Stratification Scale (CAMSIS). The CAMSIS measures occupational structure based upon social interactions [79]. Scores can range between 1 (least advantaged) and 99 (most advantaged) with a mean of 50 and a standard deviation of 15 in the population [80]. The highest score of either parent (where both were available) was used in analysis.

### 2.5.3. Parental education

Parental education qualifications have been shown to predict maths attainment trajectories in previous studies [69]. The child's parents were asked about their highest educational qualifications during

pregnancy (at 32 weeks gestation), which were coded into the following five categories: no qualifications/no higher than CSE or GCSE, vocational qualifications (i.e. teaching or nursing qualifications), O level or equivalent, A level or equivalent, and university degree. The highest qualification held by either parent (if both were available) was used in analysis; 7.3% had a CSE or below, 4.9% had a vocational qualification, 25.1% had an O level, 35.1% had an A level and 27.5% had a degree. Having a CSE or below was used as the reference group in both models.

### 2.5.4. Parent–child relationships

Parent–child relationships were included based on previous findings [69] and were evaluated using the Assessment of Mother–Child-Interaction with the Etch-a-Sketch (AMCIES) [81] (Wolke, Rios & Unzer, 1995, unpublished manuscript) task during the 'clinic in focus' sessions at age 12.5. The AMCIES involves observing parent and child dyads while they play with an Etch-a-Sketch toy. Specifically, the dyads were asked to draw a house using the Etch-a-Sketch, with either the parent or child responsible for drawing horizontal lines, and the other responsible for drawing vertical lines. To complete the task successfully, parents and their children are required to work very closely together and assist one another. Following the task, the dyads were rated by the ALSPAC team on their 'harmony', i.e. whether the relationship between them and the observed interactions were particularly negative or positive. The following 5-point scale was used to code the interactions: *many conflicts* (scored as 0), *some conflicts (generally negative with some conflict), neutral (atmosphere is neither positive or negative), quite agreeable (generally positive)* and *very agreeable (very positive and harmonious)* (scored as 4). A higher score refers to greater harmony (and a more positive relationship) between the parent and child. The AMCIES has shown good reliability in other samples (Cronbach's $\alpha = 0.76$–$0.80$; [82]).

### 2.5.5. Parental school involvement

Parental involvement in school activities was rated by the child's teacher at age 11. The activities included: 'helping in class', 'helping with out of class activities', 'attending parent-teacher sessions' and 'being involved in another school activity'. The child's teacher was asked to indicate whether the child's parents had been involved in any of these four activities by responding with *yes* or *no* to each activity, which were coded as 1 and 0, respectively. The responses for the four items were summed to create a score between 0 and 4, with a higher score indicating more parental involvement in school activities. Parental involvement in school has been found to predict maths attainment trajectories previously [69].

### 2.5.6. Working memory and IQ

Working memory (at age 10) and IQ (at age 8) were assessed during 'Clinic in Focus' sessions. Children's total IQ score was measured using the performance (short-form tests: picture completion, picture arrangement, block design and object assembly, full-form test: coding) and verbal (short-form tests: information, similarities, arithmetic, vocabulary and comprehension) subscales of the Wechsler Intelligence Scale for Children (WISC-III; [83]). The scores for each of the short-form tests were transformed to be on the same scale as though the entire test had been administered to reduce fatigue. The WISC-III holds good test–retest reliability (0.80–0.89; [84]).

Children's working memory capacity was measured using the Counting Span Task [85] administered on a computer. In this task, children are presented with screens of red and blue dots and are asked to count them out loud. After counting them correctly, children are asked to recall the number of red dots on the screens, in the same order they are presented. All screens are displayed, regardless of the child's performance. Children were shown two practice screens followed by three sets of two screens, three sets of three screens, three sets of four screens and three sets of five screens, totalling to 42 trials. The global score was used representing the number of trials children answered correctly (i.e. 0–42). Both working memory and IQ are known predictors of maths attainment [70].

### 2.5.7. Internalizing symptoms

Children's internalizing symptoms at age 11 were measured using the Strengths and Difficulties Questionnaire (SDQ; [86]), and were included in this study based on previous findings [70]. The SDQ contains 25 items assessing emotional symptoms, peer problems, conduct problems, prosocial

behaviour and hyperactivity. Parents rated their child's behaviour on each of the five subscales, with possible responses of *not true*, *somewhat true* and *certainly true*, which were coded as 0, 1 and 2, respectively, meaning that scores could range between 0 and 10 for each subscale. An 'internalizing symptoms' score was created by summing the child's scores for the emotional symptoms and peer problems subscales, with possible scores ranging from 0 to 20. A higher score is indicative of greater internalizing symptoms and problems. Example items of the internalizing symptoms scale include 'child often seems worried' and 'child is rather solitary, tends to play alone'. The SDQ overall has good concurrent and predictive validity [86], and satisfactory internal consistency (Cronbach's $\alpha$ for emotional difficulties = 0.66, and for peer problems $\alpha$ = 0.53; [87].

## 2.6. Data analysis

### 2.6.1. Exclusions and missing data

This study follows the same exclusion criteria as in Evans *et al*. [70] and Evans & Field [69]. The initial cohort was formed of 13 988 children alive at 1 year. Additional recruitment resulted in 14 901 children alive at 1 year (including singletons and twins; triplets and quadruplets were excluded due to rarity). Withdrawal from the study led to a sample size of 14 684. Data from singletons and the first-born twin were retained for analysis ($N$ = 14 498). Fourteen children were excluded as their first, or second main language was not English ($N$ = 14 484). Two thousand six hundred and fifty-two children reported to have SEN (identified by teachers at ages 7–8 and 10–11) were excluded ($N$ = 11 832). Where 50% or more of the data for the predictor variables were missing, 5342 participants were excluded, leaving a final sample size of 6490 (none of which were complete cases).

To address the issue of high attrition rates and missing data in the ALSPAC dataset (for missing data per variable, see table 1), multiple imputation was performed in R [88] using the *semTools* [89] and *Amelia* packages [90]. Eighty imputations were performed and the results were pooled [91]. The outcome variables (maths attainment KS1–KS4) were included in the imputation model but were not imputed. Instead, to address the missing outcome data, full information maximum likelihood (FIML) estimation was used [92].

### 2.6.2. Statistical analysis strategy

All analyses were conducted in R v. 3.4.3 [88]. Two latent growth models predicting maths attainment trajectories in primary and secondary education were fit using the *lavaan* package [93], which are described in more detail below.

### 2.6.3. Primary education model

The primary education model evaluates the possible effects of variables measured in primary education, and whether these predict maths attainment trajectories across the transition from primary to secondary education. The variables entered into the primary education model were as follows: school affect, student–teacher relationships, maths attitudes (age 10), teacher affect, teacher CCEI, teacher self-esteem, parental school support, child's sex, internalizing symptoms, IQ, working memory, SES, parental education and parent–child relationships. These predictors were included as exogenous observed variables that predict the intercept and slope of growth in maths attainment.

Maths attainment at 7, 11, 14 and 16 years were endogenous observed variables predicted from latent variables representing the intercept and slope for growth in maths attainment over time. The loadings for the paths from the slope latent variable to the four maths attainment outcomes were constrained to be −4 (maths at age 7), 0 (maths at age 11), 3 (maths at age 14) and 5 (maths at age 16) so that the intercept represented maths attainment just prior to the school transition at age 11 (figure 1).

### 2.6.4. Secondary education model

The secondary education model focuses on the variables measured in secondary education, and whether these predict maths attainment trajectories following the transition from primary to secondary education. The variables entered into the secondary education model were as follows: school belonging, student–teacher relationships, negative emotions towards school, maths attitudes (age 14), positive maths teaching practices, maths teacher fairness, child's sex, internalizing symptoms, IQ, working memory,

**Table 1.** Summary statistics for the key study measures. P, measured in primary education; S, measured in secondary education; WM, working memory; SDQ, internalizing symptoms; S–T, student–teacher; KS, key stage; MD, missing data.

| measure | $n$ | min | max | mdn | $M$ | 95% CI | $s$ | % MD |
|---|---|---|---|---|---|---|---|---|
| school affect (P) | 5412 | 0.00 | 24.00 | 21.00 | 20.18 | [20.09, 20.27] | 11.38 | 17% |
| S–T relationships (P) | 5716 | 0.00 | 9.00 | 8.00 | 7.52 | [7.48, 7.56] | 2.82 | 12% |
| maths attitudes (P) | 5390 | 0.00 | 40.00 | 32.00 | 29.35 | [29.08, 29.63] | 105.76 | 17% |
| teacher affect (P) | 3631 | 5.00 | 24.00 | 20.00 | 19.43 | [19.32, 19.55] | 12.91 | 44% |
| teacher CCEI (P) | 3698 | 0.00 | 38.00 | 11.00 | 12.97 | [12.72, 13.22] | 60.54 | 43% |
| teacher self-esteem (P) | 3679 | 17.00 | 40.00 | 33.00 | 32.27 | [32.10, 32.45] | 28.82 | 43% |
| school belonging (S) | 3069 | 0.00 | 18.00 | 12.00 | 12.45 | [12.35, 12.55] | 7.90 | 53% |
| S–T relationships (S) | 4816 | 0.00 | 6.00 | 4.00 | 4.07 | [4.04, 4.10] | 1.29 | 26% |
| negative emotion (S) | 3970 | 0.00 | 9.00 | 7.00 | 6.81 | [6.76, 6.86] | 2.36 | 39% |
| maths attitudes (S) | 5404 | 0.00 | 12.00 | 8.00 | 8.25 | [8.19, 8.31] | 4.97 | 17% |
| positive teaching (S) | 5317 | 0.00 | 48.00 | 34.00 | 32.76 | [32.55, 32.98] | 63.72 | 18% |
| teacher fairness (S) | 5218 | 0.00 | 20.00 | 13.00 | 12.96 | [12.84, 13.08] | 19.52 | 20% |
| SES | 5135 | 26.31 | 95.70 | 58.40 | 59.65 | [59.33, 59.97] | 137.24 | 21% |
| IQ | 5185 | 49.00 | 151.00 | 107.00 | 107.19 | [106.77, 107.60] | 237.15 | 20% |
| WM | 5115 | 0.00 | 42.00 | 19.00 | 19.32 | [19.11, 19.53] | 57.39 | 21% |
| SDQ | 5489 | 0.00 | 20.00 | 2.00 | 2.37 | [2.30, 2.44] | 6.46 | 15% |
| parent–child harmony | 5091 | 0.00 | 4.00 | 3.00 | 3.24 | [3.22, 3.26] | 0.63 | 22% |
| school support | 3770 | 0.00 | 4.00 | 1.00 | 1.78 | [1.74, 1.81] | 1.15 | 42% |
| KS1 maths | 4961 | 0.00 | 3.00 | 2.00 | 2.32 | [2.31, 2.34] | 0.28 | 24% |
| KS2 maths | 5476 | 1.00 | 6.00 | 4.00 | 4.37 | [4.35, 4.39] | 0.44 | 16% |
| KS3 maths | 4713 | 1.00 | 8.00 | 6.00 | 6.35 | [6.31, 6.38] | 1.24 | 27% |
| KS4 maths | 5137 | 2.00 | 10.00 | 8.00 | 7.50 | [7.45, 7.54] | 2.29 | 21% |

SES, parental education and parent–child relationships. As with the primary education model, these predictors were included as exogenous observed variables that predict the intercept and slope of growth in maths attainment.

The same measures of maths attainment at 7, 11, 14 and 16 years were used as endogenous observed variables predicted from latent variables representing the intercept and slope for growth in maths attainment. The loadings for the paths from the slope latent variable to the four maths attainment outcomes were constrained to be −7 (maths at age 7), −3 (maths at age 11), 0 (maths at age 14) and 2 (maths at age 16) so that the intercept represented maths attainment following the transition to secondary education at age 14 (figure 2).

For both models, all predictors were entered simultaneously. Correlation coefficients for the variables are displayed in table 2 (primary education variables) and table 3 (secondary education variables). SES, IQ and working memory were all centred prior to analysis as there was no meaningful zero in these measures. For the primary education model, scores for school affect, teacher affect, student–teacher relationships and teacher self-esteem were centred. For the secondary education model, scores for school belonging, student–teacher relationships, negative emotions towards school, positive maths teaching practices and maths teacher fairness were centred.

Two previous studies [69,70] found working memory, internalizing symptoms, parental school support, parental education and a positive parent–child relationship significantly predicted maths attainment trajectories in this sample. Due to these findings, these predictors were also included in the present analysis to adjust for their effects.

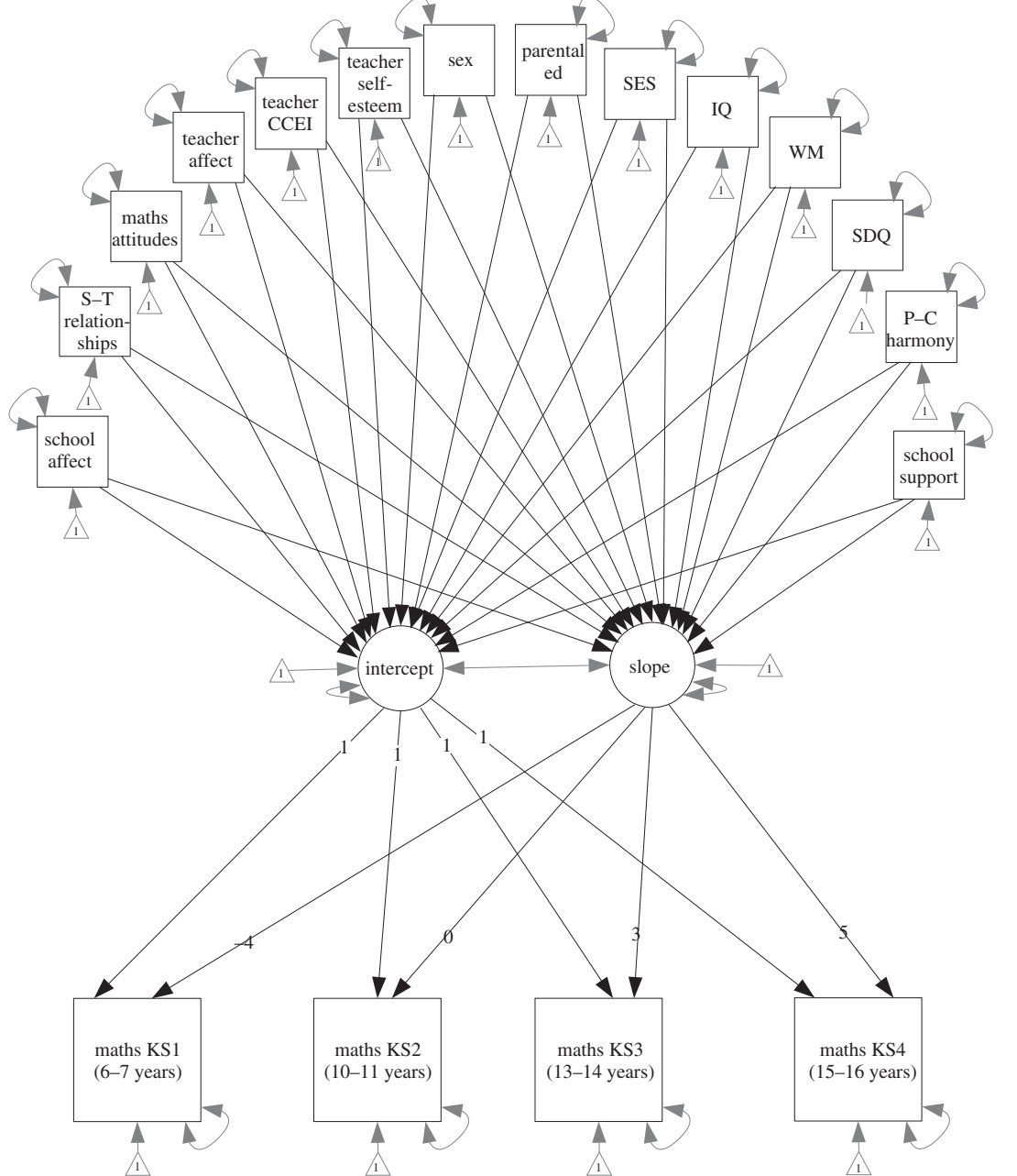

**Figure 1.** Latent growth model for maths attainment trajectories in primary education. The intercept represents maths attainment at age 11, and the slope represents maths attainment from age 7 to 16. Paths between predictor variables are implied but not illustrated. WM, working memory; SDQ, internalizing symptoms; S–T, student–teacher; ed, education; P–C, parent–child.

## 3. Results

### 3.1. Descriptive statistics

Summary statistics for the variables in the primary and secondary education models are in table 1. Maths grades were generally in line with national guidelines and expectations for all key stages. Children are expected to progress by half a grade each year in schools following the national curriculum in the UK. For both the primary education model (0.49 grades per year on average) and the secondary education model (0.46 grades per year on average), children's average growth in attainment per year was consistent with the wider population.

Both models provided satisfactory fit indices (primary education model: CFI = 0.936, TLI = 0.876, RMSEA = 0.104 [90% CI = 0.100, 0.107], SRMR = 0.05; secondary education model: CFI = 0.936, TLI = 0.876, RMSEA = 0.104 [90% CI = 0.101, 0.107], SRMR = 0.05).

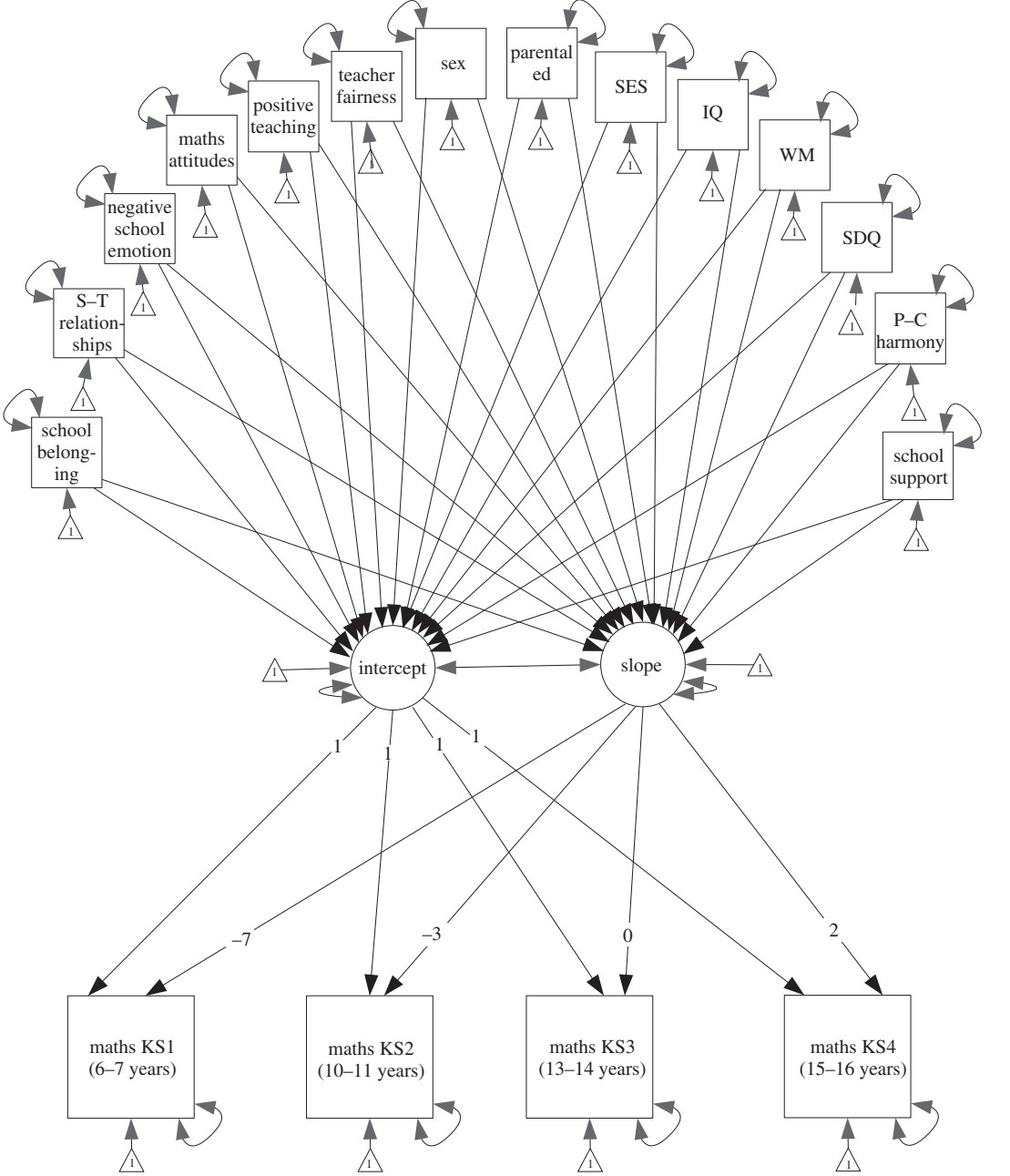

**Figure 2.** Latent growth model for maths attainment trajectories in secondary education. The intercept represents maths attainment at age 14, and the slope represents maths attainment from age 7 to 16. Paths between predictor variables are implied but not illustrated. WM, working memory; SDQ, internalizing symptoms; S–T, student–teacher; ed, education; P–C, parent–child.

## 3.2. Primary education model

### 3.2.1. Predictors of maths attainment at age 11 (intercept)

Table 4 shows the model parameters for the intercept of the primary education model. Of the substantive predictors, the only variable that significantly predicted maths attainment at the intercept was attitudes towards maths at age 10 ($p < 0.001$). School affect, student–teacher relationships, teacher-rated affect, teacher CCEI and teacher self-esteem did not significantly predict maths attainment (table 4). Of the contextual predictors, as expected, sex, parental education, SES, IQ, WM, internalizing symptoms (SDQ) and parental school support all significantly predicted maths attainment at age 11.

Maths attitudes in primary education could range between 0 and 40 with a higher score indicating more positive attitudes towards maths. The effect size of maths attitudes on maths attainment in primary education was 0.012, meaning that an increase on the maths attitudes scale by 1 point equates to an increase in maths

**Table 2.** Correlation matrix of the primary education measures and the contextual variables. The upper triangle displays the correlation coefficients and the lower triangle displays the $p$-values. S–T, student–teacher; SES, socioeconomic status; WM, working memory; SDQ, internalizing symptoms; P–C, parent–child; KS, key stage.

| variable | M | s.d. | 1 | 2 | 3 | 4 | 5 | 6 | 7 | 8 | 9 | 10 | 11 | 12 | 13 | 14 | 15 | 16 |
|---|---|---|---|---|---|---|---|---|---|---|---|---|---|---|---|---|---|---|
| 1. school affect | 20.18 | 3.37 | | 0.48 | 0.16 | 0.04 | 0.00 | 0.00 | 0.01 | 0.02 | 0.04 | −0.28 | 0.02 | 0.05 | 0.05 | 0.07 | 0.04 | 0.05 |
| 2. S–T relationships | 7.52 | 1.68 | 0.00 | | 0.11 | 0.02 | −0.01 | 0.02 | −0.01 | −0.02 | 0.02 | −0.12 | 0.03 | 0.06 | 0.00 | 0.01 | 0.01 | 0.02 |
| 3. maths attitudes | 29.35 | 10.28 | 0.00 | 0.00 | | 0.03 | −0.03 | 0.02 | −0.01 | 0.14 | 0.15 | −0.10 | −0.01 | 0.05 | 0.22 | 0.29 | 0.27 | 0.22 |
| 4. teacher affect | 19.43 | 3.59 | 0.05 | 0.39 | 0.14 | | −0.43 | 0.34 | 0.03 | 0.02 | 0.03 | 0.01 | −0.01 | 0.07 | 0.01 | 0.02 | −0.01 | 0.00 |
| 5. teacher CCEI | 12.97 | 7.78 | 0.90 | 0.55 | 0.10 | 0.00 | | −0.49 | −0.06 | −0.03 | −0.04 | −0.01 | 0.01 | −0.02 | −0.03 | −0.05 | −0.02 | −0.02 |
| 6. teacher self-esteem | 32.27 | 5.37 | 0.83 | 0.22 | 0.36 | 0.00 | 0.00 | | 0.04 | 0.00 | −0.02 | 0.00 | −0.01 | 0.04 | 0.01 | 0.01 | 0.00 | 0.00 |
| 7. SES | 59.65 | 11.72 | 0.37 | 0.36 | 0.37 | 0.12 | 0.00 | 0.04 | | 0.28 | 0.16 | −0.05 | 0.02 | 0.15 | 0.17 | 0.25 | 0.30 | 0.32 |
| 8. IQ | 107.19 | 15.40 | 0.19 | 0.32 | 0.00 | 0.39 | 0.00 | 0.95 | 0.00 | | 0.34 | −0.09 | 0.07 | 0.14 | 0.47 | 0.57 | 0.64 | 0.58 |
| 9. WM | 19.32 | 7.58 | 0.01 | 0.20 | 0.00 | 0.16 | 0.05 | 0.35 | 0.00 | 0.00 | | −0.06 | 0.02 | 0.08 | 0.26 | 0.34 | 0.36 | 0.32 |
| 10. SDQ | 2.37 | 2.54 | 0.00 | 0.00 | 0.00 | 0.45 | 0.46 | 0.83 | 0.00 | 0.00 | 0.00 | | 0.66 | 0.01 | −0.09 | −0.13 | −0.12 | −0.11 |
| 11. P–C harmony | 3.24 | 0.79 | 0.30 | 0.02 | 0.49 | 0.72 | 0.50 | 0.48 | 0.11 | 0.00 | 0.13 | 0.66 | | 0.58 | 0.09 | 0.05 | 0.11 | 0.09 |
| 12. school support | 1.78 | 1.07 | 0.00 | 0.00 | 0.01 | 0.00 | 0.29 | 0.02 | 0.00 | 0.00 | 0.00 | 0.01 | 0.58 | | 0.11 | 0.14 | 0.18 | 0.20 |
| 13. KS1 maths | 2.32 | 0.53 | 0.00 | 0.78 | 0.00 | 0.59 | 0.06 | 0.76 | 0.00 | 0.00 | 0.00 | 0.00 | 0.00 | 0.00 | | 0.54 | 0.56 | 0.50 |
| 14. KS2 maths | 4.37 | 0.67 | 0.00 | 0.44 | 0.00 | 0.24 | 0.00 | 0.67 | 0.00 | 0.00 | 0.00 | 0.00 | 0.00 | 0.00 | 0.00 | | 0.76 | 0.69 |
| 15. KS3 maths | 6.35 | 1.11 | 0.01 | 0.68 | 0.00 | 0.76 | 0.31 | 0.79 | 0.00 | 0.00 | 0.00 | 0.00 | 0.00 | 0.00 | 0.00 | 0.00 | | 0.85 |
| 16. KS4 maths | 7.50 | 1.51 | 0.00 | 0.28 | 0.00 | 0.86 | 0.24 | 0.95 | 0.00 | 0.00 | 0.00 | 0.00 | 0.00 | 0.00 | 0.00 | 0.00 | 0.00 | |

**Table 3.** Correlation matrix of the secondary education measures and the contextual variables. The upper triangle displays the correlation coefficients and the lower triangle displays the $p$-values. S–T, student–teacher; SES, socioeconomic status; WM, working memory; SDQ, internalizing symptoms; P–C, parent–child; KS, key stage.

| variable | M | s.d. | 1 | 2 | 3 | 4 | 5 | 6 | 7 | 8 | 9 | 10 | 11 | 12 | 13 | 14 | 15 | 16 |
|---|---|---|---|---|---|---|---|---|---|---|---|---|---|---|---|---|---|---|
| 1. school belonging | 12.45 | 2.81 | | 0.39 | 0.52 | 0.18 | 0.22 | 0.18 | 0.02 | −0.04 | 0.04 | −0.17 | 0.00 | 0.00 | 0.00 | 0.00 | −0.06 | −0.03 |
| 2. S–T relationships | 4.07 | 1.14 | 0.00 | | 0.34 | 0.19 | 0.26 | 0.31 | 0.07 | 0.10 | 0.07 | −0.06 | 0.04 | 0.08 | 0.09 | 0.11 | 0.17 | 0.21 |
| 3. school emotion | 6.81 | 1.53 | 0.00 | 0.00 | | 0.10 | 0.13 | 0.16 | −0.07 | −0.10 | 0.00 | −0.21 | 0.00 | −0.01 | −0.01 | 0.00 | −0.01 | −0.01 |
| 4. maths attitudes | 8.25 | 2.23 | 0.00 | 0.00 | 0.00 | | 0.45 | 0.30 | 0.00 | 0.06 | 0.06 | −0.04 | 0.02 | 0.02 | 0.11 | 0.15 | 0.21 | 0.20 |
| 5. positive teaching | 32.76 | 7.98 | 0.00 | 0.00 | 0.00 | 0.00 | | 0.62 | 0.00 | 0.04 | 0.01 | −0.04 | 0.02 | 0.00 | 0.06 | 0.07 | 0.12 | 0.11 |
| 6. teacher fairness | 12.96 | 4.42 | 0.00 | 0.00 | 0.00 | 0.00 | 0.00 | | 0.06 | 0.09 | 0.02 | −0.05 | 0.03 | 0.02 | 0.07 | 0.09 | 0.15 | 0.17 |
| 7. SES | 59.65 | 11.72 | 0.34 | 0.00 | 0.00 | 0.75 | 0.73 | 0.00 | | 0.28 | 0.16 | −0.05 | 0.02 | 0.15 | 0.17 | 0.25 | 0.30 | 0.32 |
| 8. IQ | 107.19 | 15.40 | 0.08 | 0.00 | 0.00 | 0.00 | 0.02 | 0.00 | 0.00 | | 0.34 | −0.09 | 0.07 | 0.14 | 0.47 | 0.57 | 0.64 | 0.58 |
| 9. WM | 19.32 | 7.58 | 0.07 | 0.00 | 0.82 | 0.00 | 0.43 | 0.24 | 0.00 | 0.00 | | −0.06 | 0.02 | 0.08 | 0.26 | 0.34 | 0.36 | 0.32 |
| 10. SDQ | 2.37 | 2.54 | 0.00 | 0.00 | 0.00 | 0.01 | 0.01 | 0.00 | 0.00 | 0.00 | 0.00 | | −0.01 | −0.04 | −0.09 | −0.13 | −0.12 | −0.11 |
| 11. P–C harmony | 3.24 | 0.79 | 1.00 | 0.01 | 0.77 | 0.32 | 0.30 | 0.05 | 0.11 | 0.00 | 0.13 | 0.66 | | 0.01 | 0.09 | 0.05 | 0.11 | 0.09 |
| 12. school support | 1.78 | 1.07 | 0.84 | 0.00 | 0.53 | 0.23 | 0.89 | 0.23 | 0.00 | 0.00 | 0.00 | 0.01 | 0.58 | | 0.11 | 0.14 | 0.18 | 0.20 |
| 13. KS1 maths | 2.32 | 0.53 | 0.88 | 0.00 | 0.63 | 0.00 | 0.00 | 0.00 | 0.00 | 0.00 | 0.00 | 0.00 | 0.00 | 0.00 | | 0.54 | 0.56 | 0.50 |
| 14. KS2 maths | 4.37 | 0.67 | 0.88 | 0.00 | 0.84 | 0.00 | 0.00 | 0.00 | 0.00 | 0.00 | 0.00 | 0.00 | 0.00 | 0.00 | 0.00 | | 0.76 | 0.69 |
| 15. KS3 maths | 6.35 | 1.11 | 0.01 | 0.00 | 0.76 | 0.00 | 0.00 | 0.00 | 0.00 | 0.00 | 0.00 | 0.00 | 0.00 | 0.00 | 0.00 | 0.00 | | 0.85 |
| 16. KS4 maths | 7.50 | 1.51 | 0.16 | 0.00 | 0.48 | 0.00 | 0.00 | 0.00 | 0.00 | 0.00 | 0.00 | 0.00 | 0.00 | 0.00 | 0.00 | 0.00 | 0.00 | |

**Table 4.** Model parameters for predictors of the intercept of maths attainment in primary education (age 11). $\beta$ is the standardized parameter estimate.

| predictor | b | 95% CI | $\beta$ | p-value |
|---|---|---|---|---|
| school affect | 0.001 | [−0.004, 0.007] | 0.007 | 0.643 |
| S−T relationships | −0.001 | [−0.012, 0.010] | −0.003 | 0.810 |
| maths attitudes | 0.012 | [0.010, 0.013] | 0.173 | 0.000 |
| teacher affect | −0.003 | [−0.008, 0.002] | −0.017 | 0.223 |
| teacher CCEI | −0.002 | [−0.004, 0.001] | −0.018 | 0.230 |
| teacher self-esteem | −0.001 | [−0.005, 0.002] | −0.009 | 0.535 |
| sex | 0.049 | [0.015, 0.082] | 0.036 | 0.004 |
| Edu: CSE versus vocational | −0.092 | [−0.174, −0.010] | −0.030 | 0.028 |
| Edu: CSE versus O level | 0.052 | [−0.006, 0.110] | 0.033 | 0.078 |
| Edu: CSE versus A level | 0.132 | [0.075, 0.189] | 0.092 | 0.000 |
| Edu: CSE versus degree | 0.271 | [0.206, 0.336] | 0.178 | 0.000 |
| SES | 0.004 | [0.003, 0.006] | 0.076 | 0.000 |
| IQ | 0.020 | [0.019, 0.021] | 0.450 | 0.000 |
| WM | 0.012 | [0.010, 0.014] | 0.133 | 0.000 |
| SDQ | −0.013 | [−0.020, −0.006] | −0.049 | 0.000 |
| parent–child harmony | 0.038 | [0.017, 0.059] | 0.044 | 0.000 |
| school support | 0.025 | [0.009, 0.041] | 0.039 | 0.002 |

attainment at age 11 by 0.012 levels. When looking at this effect in context, this means that the difference between children with the most positive attitudes to maths (i.e. those scoring 40), compared with those with the lowest score (i.e. those scoring 0), the difference in attainment in primary education would be the equivalent to almost a year's progress in maths ($40 \times 0.012 = 0.48$).

When looking at the contextual predictors, the results generally replicated previous findings; males were found to have slightly higher maths attainment, children to parents with a degree or A level had higher maths attainment compared with children to parents with a CSE or below, parents to children with vocational qualifications had lower attainment. Increased internalizing symptoms predicted lower attainment, and higher SES, IQ, working memory, parental school involvement and increased parent–child harmony all predicted higher maths attainment at age 11 (table 4). For a detailed discussion of these findings, see Evans *et al.* [70] and Evans & Field [69].

### 3.2.2. Primary education predictors of the rate of change

Table 5 shows the model parameters for the slope of the primary education model (i.e. the rate of change (ROC) over time). Of the substantive predictors, school affect, student–teacher relationships, teacher-rated affect, teacher CCEI and teacher self-esteem did not significantly predict maths attainment growth. Maths attitudes significantly predicted the slope of maths attainment, with more positive attitudes linked to an increased ROC over time ($b = 0.001$, $p < 0.001$). However, this effect is extremely small—when comparing children with the most positive attitudes with children with the most negative, the associated difference in attainment per year is around 0.04 for the maths-positive students.

Of the contextual predictors, the significant predictors of an increased slope for maths attainment were parental education (for those with a degree or A level), and higher SES, IQ, working memory, parental school involvement and increased parent–child harmony. Increased internalising symptoms were associated with a decreased ROC (see [70]).

Overall, the results for the primary education model suggest that the most important substantive predictor of maths attainment at age 11, and of the ROC over time is attitudes towards maths, with general school affect and teacher characteristics lacking a substantial effect on maths attainment in primary education.

**Table 5.** Model parameters for predictors of the slope of maths attainment in primary education. $\beta$ is the standardized parameter estimate.

| predictor | b | 95% CI | $\beta$ | p-value |
|---|---|---|---|---|
| school affect | 0.000 | [−0.001, 0.001] | −0.005 | 0.783 |
| S–T relationships | 0.001 | [−0.002, 0.003] | 0.012 | 0.509 |
| maths attitudes | 0.001 | [0.001, 0.002] | 0.118 | 0.000 |
| teacher affect | −0.001 | [−0.002, 0.000] | −0.025 | 0.151 |
| teacher CCEI | 0.000 | [−0.001, 0.000] | −0.012 | 0.514 |
| teacher self-esteem | 0.000 | [−0.001, 0.001] | −0.005 | 0.765 |
| sex | 0.005 | [−0.002, 0.012] | 0.021 | 0.197 |
| Edu: CSE versus vocational | −0.014 | [−0.032, 0.003] | −0.029 | 0.098 |
| Edu: CSE versus O level | 0.008 | [−0.005, 0.020] | 0.029 | 0.221 |
| Edu: CSE versus A level | 0.033 | [0.021, 0.045] | 0.140 | 0.000 |
| Edu: CSE versus degree | 0.069 | [0.055, 0.083] | 0.273 | 0.000 |
| SES | 0.001 | [0.001, 0.001] | 0.096 | 0.000 |
| IQ | 0.003 | [0.002, 0.003] | 0.364 | 0.000 |
| WM | 0.002 | [0.001, 0.002] | 0.109 | 0.000 |
| SDQ | −0.002 | [−0.003, −0.000] | −0.041 | 0.012 |
| parent–child harmony | 0.006 | [0.001, 0.010] | 0.039 | 0.014 |
| school support | 0.005 | [0.002, 0.009] | 0.051 | 0.002 |

## 3.3. Secondary education model

### 3.3.1. Predictors of maths attainment at age 14 (intercept)

The parameters for the secondary education model are reported in table 6. The statistically significant substantive predictors of maths attainment at age 14 were school belonging ($p = 0.001$), student–teacher relationships ($p < 0.001$), attitudes towards maths at age 14 ($p < 0.001$) and maths teacher fairness ($p = 0.002$). Negative emotion towards school, and positive teaching in maths did not significantly predict maths attainment in secondary education ($ps = 0.404$ and $0.118$, respectively).

Unexpectedly, school belonging was negatively associated with attainment, meaning that students reporting greater school belonging in secondary education had lower maths attainment at age 14 ($b = −0.018$); however, this effect was relatively small. Student–teacher relationships had a stronger effect on maths attainment ($b = 0.059$), whereby students rating their relationship with their teachers as more positive had higher maths attainment. When comparing the lowest scores with the highest, this difference would equate to approximately one-third of a grade increase for the students rating their student–teacher relationships as highly as possible on the scale, which is generally a small effect.

Attitudes towards maths were associated with maths attainment in secondary education, where more positive attitudes equated to increased maths attainment. Maths attitudes could range from 0 to 12, meaning that a 1-unit increase for maths attitudes on this scale equated to an increase in attainment by 0.064 national curriculum levels. When comparing the lowest-rated maths attitudes (i.e. the most negative) with the highest-rated maths attitudes (i.e. the most positive), the difference in attainment would be around 0.77 levels. Maths teacher fairness also significantly predicted maths attainment, with greater perceived fairness and equality equating to a 0.011 unit increase in attainment. However, this effect was extremely small.

When looking at the contextual predictors, all variables predicted maths attainment in regard to statistical significance (all $ps < 0.05$), with the only exception of parental education when looking at differences between children to parents with an O level compared with those with a CSE and below ($p = 0.101$). Being male, having higher SES, IQ and working memory were linked to higher attainment, greater internalizing symptoms predicted decreased attainment, and both increased parent–child harmony, and parental school support equated to increased attainment (see [69] for further discussion).

**Table 6.** Model parameters for predictors of the intercept of maths attainment in secondary education (age 14). $\beta$ is the standardized parameter estimate.

| predictor | $b$ | 95% CI | $\beta$ | $p$-value |
|---|---|---|---|---|
| school belonging | −0.018 | [−0.028, −0.007] | −0.047 | 0.001 |
| S–T relationships | 0.059 | [0.034, 0.084] | 0.066 | 0.000 |
| negative school emotion | 0.008 | [−0.011, 0.027] | 0.012 | 0.404 |
| maths attitudes | 0.064 | [0.052, 0.076] | 0.139 | 0.000 |
| positive teaching | −0.003 | [−0.007, 0.001] | −0.026 | 0.118 |
| teacher fairness | 0.011 | [0.004, 0.018] | 0.048 | 0.002 |
| sex | 0.078 | [0.029, 0.128] | 0.038 | 0.002 |
| Edu: CSE versus vocational | −0.122 | [−0.242, −0.002] | −0.027 | 0.047 |
| Edu: CSE versus O level | 0.071 | [−0.014, 0.156] | 0.030 | 0.101 |
| Edu: CSE versus A level | 0.224 | [0.140, 0.308] | 0.105 | 0.000 |
| Edu: CSE versus degree | 0.458 | [0.362, 0.554] | 0.201 | 0.000 |
| SES | 0.007 | [0.004, 0.009] | 0.076 | 0.000 |
| IQ | 0.028 | [0.026, 0.030] | 0.422 | 0.000 |
| WM | 0.019 | [0.015, 0.022] | 0.138 | 0.000 |
| SDQ | −0.022 | [−0.032, −0.012] | −0.055 | 0.000 |
| parent–child harmony | 0.049 | [0.019, 0.080] | 0.038 | 0.002 |
| school support | 0.040 | [0.017, 0.063] | 0.042 | 0.001 |

### 3.3.2. Secondary education predictors of the rate of change

Table 7 shows the model parameters for the secondary education ROC in maths attainment. Of the substantive predictors, school belonging, student–teacher relationships, maths attitudes and maths teacher fairness, all significantly predicted the slope of maths attainment (all $p$s < 0.001). Negative emotion towards school, and positive teaching in maths did not significantly predict growth in maths attainment (table 7).

Consistent with the intercept, greater school belonging predicted a slower ROC in maths attainment, whereby a 1 unit increase in school belonging equated to a decrease in the ROC by 0.003. However, this effect is extremely small, given that the average change in attainment was 0.46 grade levels per year. Similarly, the effects of student–teacher relationships and maths teacher fairness were also particularly small, with more positive student–teacher relationships and increased teacher fairness associated with an increased ROC by 0.009 and 0.002, respectively. More positive maths attitudes were linked to an increased ROC ($b = 0.008$), suggesting that adolescents with a more positive attitude towards maths at age 14 progressed at a quicker rate, though ultimately, this is a small effect.

Of the contextual predictors, there were no significant differences in the ROC between male and female students. Children to parents with a degree or A level had an increased ROC. Higher SES, IQ and working memory equated to an increased ROC. Increased internalizing symptoms equated to a slower ROC, and greater parent–child harmony and parental school involvement predicted a faster ROC (see [69]).

Generally, the results for the secondary education model suggest that there are aspects of the secondary school environment that are important for maths attainment trajectories within secondary education, and that there are also child-specific factors, specifically their attitudes towards maths, that have strong associations with maths attainment; however, broadly the effects on attainment were quite small.

## 4. Discussion

The aim of this study was to explore predictors of maths attainment trajectories in primary and secondary education by focusing specifically on the school climate and children's affect towards

**Table 7.** Model parameters for predictors of the slope of maths attainment in secondary education. $\beta$ is the standardized parameter estimate.

| predictor | b | 95% CI | $\beta$ | p-value |
|---|---|---|---|---|
| school belonging | −0.003 | [−0.004, −0.001] | −0.061 | 0.001 |
| S–T relationships | 0.009 | [0.006, 0.013] | 0.093 | 0.000 |
| negative school emotion | 0.001 | [−0.002, 0.003] | 0.010 | 0.600 |
| maths attitudes | 0.008 | [0.006, 0.009] | 0.148 | 0.000 |
| positive teaching | 0.000 | [−0.001, 0.000] | −0.030 | 0.150 |
| teacher fairness | 0.002 | [0.001, 0.003] | 0.068 | 0.001 |
| sex | 0.005 | [−0.002, 0.012] | 0.022 | 0.162 |
| Edu: CSE versus vocational | −0.013 | [−0.030, 0.003] | −0.027 | 0.116 |
| Edu: CSE versus O level | 0.007 | [−0.004, 0.019] | 0.029 | 0.220 |
| Edu: CSE versus A level | 0.033 | [0.021, 0.044] | 0.138 | 0.000 |
| Edu: CSE versus degree | 0.067 | [0.053, 0.080] | 0.264 | 0.000 |
| SES | 0.001 | [0.001, 0.001] | 0.090 | 0.000 |
| IQ | 0.003 | [0.002, 0.003] | 0.357 | 0.000 |
| WM | 0.002 | [0.001, 0.002] | 0.118 | 0.000 |
| SDQ | −0.002 | [−0.003, −0.001] | −0.046 | 0.003 |
| parent–child harmony | 0.005 | [0.001, 0.009] | 0.035 | 0.023 |
| school support | 0.005 | [0.002, 0.008] | 0.047 | 0.003 |

school, student–teacher relationships, teacher characteristics, attitudes towards maths and perceptions of the maths teacher.

## 4.1. Summary of main results

The primary education model investigated the associations between maths attainment trajectories of adolescents and their affect towards school, perceived student–teacher relationships, attitudes towards maths and characteristics of their teacher (affect towards teaching, mental wellbeing and self-esteem) in primary education, while adjusting for known predictors and demographic variables. The only statistically significant predictor of maths attainment was children's attitudes towards maths, where more positive attitudes towards maths predicted increased attainment at age 11, and an increased ROC over time. The magnitude of the effect of maths attitudes for attainment at age 11 was moderate; a 10 unit increase on the maths attitudes scale equated to an increase in attainment by 0.12 national curriculum levels. When comparing children with the lowest score for maths attitudes (0), with the highest (40), this difference would be close to a year's worth of progress (i.e. almost half a grade level). The size of the effect on yearly progress was small; a 10 unit increase on the maths attitude scale equates to an increased ROC of 0.01 grade levels per year. Affect towards school, student–teacher relationships and teacher characteristics were not found to significantly predict maths attainment at age 11, nor the ROC.

The secondary education model examined school belonging, negative emotion towards school, relationships with teachers, attitudes towards maths and perceptions of the maths teacher (positive teaching practices and fairness) in secondary education as predictors of maths attainment after adjusting for known predictors and demographic variables. School belonging, student–teacher relationships, maths attitudes and maths teacher fairness were significantly associated with maths attainment at age 14, and the ROC. Unsurprisingly, student–teacher relationships rated as more positive and greater maths teacher fairness were associated with increased attainment trajectories, though the effects were relatively small. Maths attitudes were positively associated with maths attainment at age 14 and an increased ROC over time with a considerable effect size. School belonging was negatively associated with maths attainment, meaning that increased school belonging was linked

to decreased maths attainment at age 14, and a slower ROC over time. Positive teaching practices in maths and negative emotion towards school were not significantly associated with maths attainment.

Based on the wider literature, it was expected that positive attitudes towards maths would be associated with increased attainment in both school environments, which was supported by the results with moderately large effect sizes present in both models. The findings suggest that children who enjoy maths and perceive it to be useful, interesting and important achieve higher grades than their peers who feel more negatively about maths. However, it is important to note that this result does not imply causality. It could be that enjoying maths increases grades through greater motivation, practice and effort, but also, that feeling competent in maths and achieving good grades increases enjoyment. It is highly likely that there is a reciprocal relationship where attitudes affect achievement, and achievement affects attitudes, which has been found in existing research [94]; however, this idea could not be examined in this study.

When looking at the findings for the secondary education model, it appears that school-related factors in secondary education have a greater effect on maths attainment trajectories, where more positive student–teacher relationships, greater perceived maths teacher fairness and lower school-belonging were significantly associated with increased attainment. It was expected that positive student–teacher relationships and teacher fairness would be positively associated with attainment. However, it is somewhat surprising that student-reported school-belonging was negatively associated with attainment. It appears that the school-climate declines around the transition [45]; however, this still does not explain the negative association with maths as found here. One possible explanation is that the measure used for school belonging in this study contained items relating to the child's peer relationships, and as such, could reflect their perceived popularity. For example, items for the school belonging composite included 'my school is a place where I know people who think a lot of me', 'my school is a place where I get on well with other pupils in my classes' and 'my school is a place where other pupils are very friendly'. Therefore, it could be that adolescents who perceive their peers as more accepting and friendly, are those with a greater number of friendships and are considered 'popular', which has been associated with decreased attainment [95]. Another potential explanation could be that students who are especially 'gifted' in maths may not feel comfortable socially within their school or may not find it sufficiently challenging intellectually. Peer victimization is high for gifted students [96], and so it could be that high-achieving maths students do not view their school as a place they get on well with other pupils. In addition, stronger mathematicians may feel less engaged by classwork they do not find particularly challenging, and so may not identify strongly with their school and their educational environment. However, additional research is needed to investigate these possibilities further.

The findings here support the idea that positive student–teacher relationships are an important part of the school climate associated with long-term positive outcomes. This measure generally focused on *all* teachers students interacted with. However, when focused on the adolescent's maths teacher specifically, this study found support for teacher fairness as a predictor of maths attainment trajectories, but not for positive teaching practices (relating to the perceived efficacy of the teacher, their encouragement and their emphasis on the importance of effort). The significant finding of teacher fairness suggests that students who perceive their maths teacher as treating all students equally (regardless of gender or ability) had increased maths attainment trajectories. This is supported by existing research particularly on the damaging effects of gender stereotypes in maths (e.g. [58]), and further demonstrates the importance of treating students equally, regardless of their characteristics and abilities.

Positive teaching practices (i.e. the teacher tries to make maths interesting, tells the class why maths is important and understands maths really well) were not found to predict maths attainment significantly. This finding implies that the perceived competence of the maths teacher is not associated with students' maths attainment, and other teacher-related factors (such as teacher fairness) are more important in secondary education. This finding is unexpected; however, one possible explanation for the absence of a significant finding could be that at average levels of teacher fairness (in the model, this variable was centred), the instructional quality of teachers is less important. It could be that when students perceive themselves to be viewed equally, they are less likely to become disengaged with difficult work, regardless of their abilities and the competency of their teacher, provided that they are treated similarly to their peers. However, further analyses and research would be needed to explore this idea.

Negative emotion towards school, including feelings of loneliness, worry and restlessness, was not found to significantly predict maths attainment. One possible explanation for the absence of a significant association could be that negative emotion towards school was measured at only one specific timepoint, meaning that any changes in affect towards school would not be accounted for. To

illustrate, adolescents could be experiencing short-term but heightened stress and emotion towards school relating to exams or assessments, the breakdown of friendship groups, or issues with bullying and victimization. Their feelings towards any school-related short-term stressors may have been reflected in their responses to the questionnaire, but may not have been long-lasting enough to affect their overall attainment. However, this idea can only be speculated as multiple measures of emotion towards school over time were not available.

When looking at the results of the models together, there are several interesting findings. Firstly, the absence of a significant association between school- and teacher-related variables in primary education and maths attainment is surprising. Based on the existing literature, it was predicted that a positive school climate, a warm student–teacher relationship and positive teacher characteristics (i.e. positive affect towards teaching, high self-esteem and fewer mental health symptoms) in primary education would be associated with increased maths attainment. These findings suggest that the effects associated with poor secondary education experiences could be more substantial than positive primary education experiences. It could be that the significant associations found in the secondary education model reflect the greater importance of the secondary education environment for attainment, or the lack of fit between adolescents' changing needs and their educational environment as proposed by the stage–environment fit theory [44]. Eccles *et al.* [44] suggest that there are fewer opportunities for positive student–teacher relationships in secondary education, especially where children transition from having one teacher per year in primary education to interacting with multiple teachers throughout the day in secondary education, which may help explain why student–teacher relationships were associated with maths attainment in secondary education but not in primary education. The school climate is also thought to differ substantially between primary and secondary education (such as a greater emphasis on discipline, social comparison and public evaluation in secondary education) which could explain why children's affect towards school was associated with maths attainment in secondary education, but not in primary education. It appears that children experiencing maladaptive transitions to secondary education, where their needs are vastly different to their environment, are potentially the most at-risk of poor attainment, which is supported by previous research [12]. Other possible explanations could be that the effects are due to the differences in measures used in both models; however, further analyses would be needed to assess this further.

## 4.2. Contextual predictors

These variables were included to adjust for known effects from previous studies. They are discussed in detail in Evans *et al.* [70] and Evans & Field [69], and so here we will briefly summarize the key points. The results suggest that males have significantly greater maths attainment at ages 11 and 14, but their rate of growth per year is not significantly different from females, implying that by early adolescence males have a slight grade-advantage. It is apparent that even when school-related factors and attitudes towards maths are adjusted for, there are still differences in maths attainment between adolescent males and females.

Unsurprisingly, greater IQ, SES and working memory predicted greater attainment at age 11 and 14, and an increased ROC over time in both models. Fewer internalizing symptoms, greater parental school support and a more positive parent–child relationship were also associated with increased maths attainment trajectories in both models. Parental education qualifications were also found to significantly predict maths attainment trajectories. In both models, when compared with children of parents with a CSE qualification or below, having a vocational qualification was associated with decreased attainment (not significant for the ROC), and having an A level or degree was associated with increased attainment. There were no significant differences between children to parents with an O level and a CSE or below for attainment at age 11 and 14 or the ROC. Overall, the findings indicate that higher levels of parental education qualifications are generally linked to increased maths attainment trajectories.

## 4.3. Implications, limitations and future research directions

Together, the findings suggest that the secondary education school environment and children's attitudes towards maths have important implications for children's maths attainment throughout school. Based on these findings, there are several recommendations for educational strategies to help improve maths attainment. We could not assess causal links in this study, but it appears that improving children's attitudes towards maths might help improve their attainment. This has been achieved by the Maths

Counts programme [97]; however, it is likely that the associated increase in maths abilities prompted by this programme also increases children's attitudes towards maths.

Focusing predominantly on the secondary education environment could also be useful when targeting children's attainment. In this study, we found that student–teacher interactions (specifically students' relationships with teachers and their maths teacher's fairness) in secondary education had a significant association with attainment. These findings imply that one potential focal area for maths interventions could involve improving these relationships and interactions. However, it is important to note that it is also possible that children underachieving in maths have worse relationships with their teachers as a result of their poor performance, i.e. where children have received harsh feedback, or have experienced unpleasant public evaluation of their low abilities, and consequently dislike their teachers.

Another key finding of this study was the negative association between school belonging in secondary education and maths attainment, which implies that high-achieving maths students may not feel particularly happy in their secondary school. Based on this idea, secondary schools could potentially help students feel more comfortable in their surroundings by providing a warmer school climate and by making adolescents' educational environment a more positive place to be.

Overall, these findings provide insight into what aspects educational interventions could potentially focus on when aiming to improve maths attainment. Future research should focus on determining the causality of these associations to better understand how these factors affect attainment, and to identify the most effective methods to improve maths outcomes.

The application and interpretation of these findings are affected by methodological limitations that warrant further discussion. Firstly, this study aimed to focus on the effects of the transition to secondary education, comparing school-related factors across the transition to secondary education and how they may affect maths attainment trajectories. However, the timings of the measures used are not directly before and following the transition, meaning that we cannot say with any certainty that the transition event itself had a direct impact on the outcomes. Additionally, the measures are not directly comparable pre- and post-transition meaning that we were unable to look at any changes over time in school-related affect, student–teacher relationships and maths attitudes. Another weakness of the study was due to the availability of measures within the ALSPAC dataset. It was not possible to include several measures that have been linked to maths outcomes in previous studies (such as teachers' attitudes towards maths and their maths anxiety for example), or to include variables that were measured in both primary and secondary education (i.e. attitudes towards maths teacher). Including these predictors would have been useful in obtaining a more comprehensive model of maths attainment trajectories, and by assessing the relative importance of different factors for attainment in primary and secondary education.

Despite the clear advantages of using a large birth cohort such as ALSPAC, including the large sample size and breadth of measures available, there are limitations to be considered relating to the generalizability of the findings. For example, adolescents within the ALSPAC sample have slightly higher grades than the population [71]. There was also very little bullying reported by parents of children in ALSPAC, which when compared with current figures for the rest of the UK [98] suggests that the ALSPAC sample had relatively positive school experiences and peer relationships. Future research would benefit from a more diverse sample as it could be that victimized children view their school-climate and student–teacher relationships differently compared with non-victimized children, thus experiencing different school affect than those in this study.

## 5. Conclusion

Overall, of all the variables analysed here, this study found that the most important school-related predictor of maths attainment trajectories in primary and secondary education was children's maths attitudes. This effect was unsurprisingly strong with gains in attainment up to half a grade level at age 11, and around three-quarters of a grade level at age 14 when comparing children with the worst-rated maths attitudes with children with the best-rated maths attitudes. There were differences between primary and secondary variables where aspects of the school climate (student–teacher relationships and school belonging) had a significant association with attainment in secondary education, but not in primary education. However, it cannot be determined from this study alone whether the differences in the predictive power of variables in primary and secondary education were due to the transition event, structural differences between primary and secondary education, other age-related changes in

development or differences between measures used in the models. Due to methodological limitations and inconsistencies within the literature, the practical applications of the findings are reduced. However, it is apparent that schools should aim to emphasize and encourage positive student–teacher relationships (particularly in secondary education where opportunities for this is reduced), develop children's positive attitudes towards maths and ensure teachers treat all students equally.

Ethics. Ethical approval for this research was granted by the University of Sussex Cross-Schools Research Ethics Committee under submission code ER/DE84/1. Ethical approval was obtained from the ALSPAC Ethics and Law Committee and the Local Research Ethics Committees. Informed consent for the use of data collected via questionnaires and clinics was obtained from participants following the recommendations of the ALSPAC Ethics and Law Committee at the time.

Data accessibility. Data used for this submission will be made available on request to the Executive (alspacexec@bristol.ac.uk). The ALSPAC data management plan (http://www.bristol.ac.uk/alspac/researchers/data-access/documents/alspac-datamanagementplan.pdf) describes in detail the policy regarding data sharing, which is through a system of managed open access. Code for analysis is available at https://osf.io/vjzad/?view_only=26b61405b5784ed4a9cd061d7518640a.

Authors' contributions. D.E. and A.P.F. conceived the study. D.E. conducted initial data processing and ran all statistical analyses. D.E. wrote the manuscript and A.P.F. reviewed and revised the manuscript and data analysis process at all stages. All authors gave final approval for publication.

Competing interests. We declare we have no competing interests.

Funding. The UK Medical Research Council and the Wellcome Trust (grant no. 102215/2/13/2) and the University of Bristol provide core support for ALSPAC. A comprehensive list of grants funding is available on the ALSPAC website (http://www.bristol.ac.uk/alspac/external/documents/grant-acknowledgements.pdf). This research was specifically funded by Department for Education and Skills (grant no. EOR/SBU/2002/121) and the Wellcome Trust and MRC (grant no. 092731). This publication is the work of the authors and they will serve as guarantors for the contents of this paper. This specific research project did not receive any funding.

Acknowledgements. The authors are extremely grateful to all the families who took part in this study, the midwives for their help in recruiting them and the whole ALSPAC team, which includes interviewers, computer and laboratory technicians, clerical workers, research scientists, volunteers, managers, receptionists and nurses.

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
