## [Reviewer comments · Royal Society Open Science]

Review History

RSOS-200975.R0 (Original submission)

Review form: Reviewer 1

Is the manuscript scientifically sound in its present form?

Yes

Are the interpretations and conclusions justified by the results?

Yes

Is the language acceptable?

Yes

Do you have any ethical concerns with this paper?

No

Have you any concerns about statistical analyses in this paper?

No

Recommendation?

Accept with minor revision (please list in comments)

Comments to the Author(s)

Please note that page numbers below refer to the pages in the pdf, which are $n + 1$ for page n in the article.

I enjoyed reading this article. I think it is well written and is a sound piece of research. I look forward to seeing it published. I have some comments and small suggestions below.

Intro – This is framed as a problem of maths attainment, and yet the statistics about few adults having skills commensurate with those of children suggests that the problem may be as much about retention over time as about the initial learning in school?

It is repeatedly stated (e.g. p. 3; p. 5 line 50) that “Children have several specialised subject teachers in secondary education compared to one individual teacher for all subjects for the entire school-year in primary education.” This is less true now in England than it was, especially in maths, where specialist maths teachers are increasingly common in primary, so this may need to be nuanced.

p. 4. There is perhaps a bit of slippage into ‘causal’ language here, with ‘negative effects’ and talk about interventions, and ‘beneficial’ (line 38), which is all carefully avoided in most places elsewhere.

p. 4 line 49 “much greater teacher to student ratio” – I think ‘lower’ is what is meant here? But I don’t think this is true. Certainly not ‘much’. Classes are capped at 30 in primary, but are rarely larger than this on average at secondary, in my experience.

p. 7, 2(a)(i) – I was surprised that SEND children were excluded. Were there not enough of these to do a separate subgroup analysis? You might find larger effects for this group, perhaps? I think schools would be interested in the findings of this.

p. 7 lines 54-55 these ages are for the END of key stage years (e.g., Year 9 for the end of KS3, but KS3 is years 7-9, so ages 11-14).

p. 7-8, 2(b)(i) Some of this information is out of date. There are no longer KS3 tests. And GCSEs are now graded on a 1-9 scale.

p. 8 line 28. I wonder if there is a reliability/validity problem here? I would avoid a word like ‘somewhat’ with young children. Do they ever use such a word or even know what it means? I could imagine some children guessing that ‘somewhat agree’ might mean ‘strongly agree’, which would mess up your scale.

p. 8 line 44. In a similar way, I am not even sure that I am clear myself about the difference between ‘partly true’ and ‘somewhat true’, so this seems like a particularly problematic choice of words for a scale. It is surprising the alpha is so high if people interpreted these differently? I suppose their presentation in order along a line meant that the words didn’t matter too much.

p. 9 line 23 – these names seem much better to me

p. 8 (c) (i) I think I need to know what the other statements were, as only a few examples are given. Otherwise, saying that ‘they’ reduced to 2 factors is a bit meaningless, as I don’t know what ‘they’ were – what variables reduced to 2 factors? Maybe all your statements just happened to be clearly about those 2 things? Could the complete lists be included in the appendices? Similarly on Line 53 you say that ‘Three factors were included’, but I think this language is a bit confusing, as this makes me think of factors emerging from a factor analysis, but these were measured using separate instruments.

p. 10 (e) (iv) Is there any reliability/validity data for this scale?

p. 18 “more positive attitudes equated to lower maths attainment” – I agree that this is very surprising. I suppose there is no possibility of a coding error or something?

bottom of p. 20 – top of p. 21 I don't think it is so surprising that the stronger mathematicians may feel less at home at school and are to some extent social 'misfits'. Regarding the subsequent discussion on p. 21, there may be findings in the PISA data to support strong maths students not liking the subject. Could it also be that stronger maths students are harsher critics of their teachers – the better you are at maths the more likely you may be to think that your teacher is a fool? 8-)

p. 22 top, maybe the stronger maths students are more likely to be picked on by other students and have a less happy experience in school?

I am just suggesting these things as they occur to me in case they help your discussion.

Small suggestions:

Consider 'mathematics' rather than 'maths' throughout

Would 'primary' and 'secondary' be simpler language for the reader than 'pre-transition' and 'post-transition'?

p. 7, 2(b)(i) – not sure why there is (i), as there is no (ii)?

p. 13 Table 1 (and all the other tables) – the header row was shaded so darkly that I could hardly read the column headings (only by zooming in enormously)

p. 13 line 46 – I prefer to avoid using words like 'very' (especially in italics) in scientific writing – can the value be stated instead? Similarly, on p. 20 line 18 I was unsure how to interpret “somewhat substantial”???

p. 16 line 23 missing apostrophe in 'years'

A couple of times you use the word “Although” where I think it is grammatically incorrect, and should be “However” (p. 16 Line 40; p. 18 line 38)

p. 17 Could the significant predictors in these tables be indicated in some way for the convenience of the reader?

Review form: Reviewer 2

Is the manuscript scientifically sound in its present form?

Yes

Are the interpretations and conclusions justified by the results?

Yes

Is the language acceptable?

Yes

Do you have any ethical concerns with this paper?

No

Have you any concerns about statistical analyses in this paper?

No

Recommendation?

Major revision is needed (please make suggestions in comments)

Comments to the Author(s)

This is an interesting manuscript with a useful and timely focus. It is very clearly structured, and is written clearly, concisely and in an engaging way. However, there are three issues that mean it is not possible to recommend publication of the manuscript in its current form. These issues are to do with the framing of the study, the definition and justification of the variables included in the analysis, and with the interpretation of some of the findings. It may be that these issues are relatively straightforward to deal with – but they seem substantial enough to recommend major, rather than minor, revisions, especially as they relate to aspects across most sections of the current manuscript.

With regard to the framing of the research study, it is not clear to what extent the article is really about transition. Much of the introduction suggests this is a key focus – including Sections 1a and 1c. However, the discussion makes clear that actually the data available do not allow the authors to say much about the point of transition – much of the data is from when children were 14 years old and had already been at their secondary school for 2-3 years. Also, Stage-Environment Fit theory is not mentioned again after the introduction, which suggests that it was not a useful tool to interpret the results of the analysis. I recommend that the emphasis on the point of transition be reduced in the introduction, and that a more consistent focus is placed on relationships between attitudes and attainment throughout the paper – including on whether we might expect relationships between attitudes and attainment to be similar or different in secondary school compared to primary school.

With regard to the measures, there is a need for some more information in some areas and some justification for the inclusion of some measures in others. More information is certainly needed about the mathematics achievement measures used. It is not possible to tell from the text in Section 2bi what variables have actually been used from the NPD. Readers will therefore not be able to know what the data represent – whether they are national standardised tests or teacher assessments, or some combination of the two. Actual variable names from the NPD should be given – e.g. for KS3 outcome – was this NC level, or paper test score(s), or teacher assessment, or a combination of these? I would think that total score on tests would be the most valid measure – i.e. KS3.KS3_MATTOTMRK – as it is clear what this represents and this variable is not subject to teacher bias in the same way as a teacher assessment of NC level may be.

A further issue with the measures is a need for clearer articulation between the literature review and the variables included in the analysis. In most cases this is clear, but there are a few where the connection is not clear. The main examples of this are the teacher characteristics included in the pre-transition model. On page 6, the manuscript states, “In addition to the effects associated with a positive student-teacher relationship, teachers’ own attitudes, self-efficacy beliefs, and abilities can influence the development of students’ attitudes towards maths regarding gender stereotypes [57], and can affect students’ attainment [57,58]. Teachers’ enjoyment of maths also affects the instructional time given to maths in that teachers who enjoy maths more spend more time engaging in maths tasks [59].” However, none of these factors seem to connect directly with teachers’ own mental health as measured with the CCEI or BSE measures. Therefore it seems that these could have just been included because they are available in the dataset – some clearer justification for their inclusion should be given.

Finally, with regard to the interpretation of some of the findings in the discussion section, there is a need for some more detailed analysis – particularly of the more unexpected findings. In the current discussion, it is not sufficiently clear to what extent the authors think that these are informative or important to research further, given the data drawn on and the literature that hypotheses were initially based on. For example, there is a difference in the effect of relationship with teachers on attainment, between 11 years old and 14 years old. When this is first discussed, in Section 4a on page 20, the text would benefit from a more thorough analysis of potential reasons why this may be. These reasons would include the possibility that it could be a genuine effect, but also that it could be due to the different measures included in the two models, or differences in structural aspects of primary versus secondary education in England (including the

fact that students generally have 1 teacher in primary school, but many teachers in secondary school). Again, clearer articulation between sections in the manuscript would help the reader in making sense of the findings – for example in the introduction, primary school teachers' attitudes to mathematics are mentioned as an important factor, but do not appear to have been included in the pre-transition model (presumably because this data is not available). This is fine, but it would help the reader if this was discussed more explicitly in the manuscript.

Some further development of the text in response to these issues would make for a much stronger contribution to the literature. The core concepts of the work are sound. In its current form, the value of the research is just not as clear as it could be.

Decision letter (RSOS-200975.R0)

Dear Ms Evans,

The editors assigned to your paper ("Maths attitudes, school affect, and teacher characteristics as predictors of maths attainment trajectories across primary to secondary education") have now received comments from reviewers. We would like you to revise your paper in accordance with the referee and Associate Editor suggestions which can be found below (not including confidential reports to the Editor). Please note this decision does not guarantee eventual acceptance.

Please submit a copy of your revised paper before 19-Aug-2020. Please note that the revision deadline will expire at 00.00am on this date. If we do not hear from you within this time then it will be assumed that the paper has been withdrawn. In exceptional circumstances, extensions may be possible if agreed with the Editorial Office in advance. We do not allow multiple rounds of revision so we urge you to make every effort to fully address all of the comments at this stage. If deemed necessary by the Editors, your manuscript will be sent back to one or more of the original reviewers for assessment. If the original reviewers are not available, we may invite new reviewers.

If your study uses humans or animals please include details of the ethical approval received, including the name of the committee that granted approval. For human studies please also detail

whether informed consent was obtained. For field studies on animals please include details of all permissions, licences and/or approvals granted to carry out the fieldwork.

- Data accessibility

If you wish to submit your supporting data or code to Dryad (<http://datadryad.org/>), or modify your current submission to dryad, please use the following link:
<http://datadryad.org/submit?journalID=RSOS&manu=RSOS-200975>

- Competing interests

- Authors' contributions

- Acknowledgements

- Funding statement

on behalf of Dr Emma Hayiou-Thomas (Associate Editor) and Essi Viding (Subject Editor)
 openscience@royalsociety.org

Associate Editor's comments (Dr Emma Hayiou-Thomas):

I enjoyed reading the third in this series of studies examining the predictors of mathematical attainment in the ALSPAC sample. Along with both reviewers, I found it to be well-designed and clearly-written, with thought-provoking findings - the changes in the relationship between school factors and attainment in primary vs secondary are really very striking. Both reviewers have made a number of constructive suggestions for how to further strengthen this paper, particularly around clarity and justification of the measures, as well as the framing of the motivations and discussion of the unexpected findings. Please address each of these points carefully - I look forward to reading the revised manuscript.

Reviewers' Comments to Author:

Reviewer: 1

Comments to the Author(s)

Please note that page numbers below refer to the pages in the pdf, which are $n + 1$ for page n in the article.

I enjoyed reading this article. I think it is well written and is a sound piece of research. I look forward to seeing it published. I have some comments and small suggestions below.

Intro - This is framed as a problem of maths attainment, and yet the statistics about few adults having skills commensurate with those of children suggests that the problem may be as much about retention over time as about the initial learning in school?

It is repeatedly stated (e.g. p. 3; p. 5 line 50) that "Children have several specialised subject teachers in secondary education compared to one individual teacher for all subjects for the entire school-year in primary education." This is less true now in England than it was, especially in maths, where specialist maths teachers are increasingly common in primary, so this may need to be nuanced.

p. 4. There is perhaps a bit of slippage into 'causal' language here, with 'negative effects' and talk about interventions, and 'beneficial' (line 38), which is all carefully avoided in most places elsewhere.

p. 4 line 49 "much greater teacher to student ratio" - I think 'lower' is what is meant here? But I don't think this is true. Certainly not 'much'. Classes are capped at 30 in primary, but are rarely larger than this on average at secondary, in my experience.

p. 7, 2(a)(i) - I was surprised that SEND children were excluded. Were there not enough of these to do a separate subgroup analysis? You might find larger effects for this group, perhaps? I think schools would be interested in the findings of this.

p. 7 lines 54-55 these ages are for the END of key stage years (e.g., Year 9 for the end of KS3, but KS3 is years 7-9, so ages 11-14).

p. 7-8, 2(b)(i) Some of this information is out of date. There are no longer KS3 tests. And GCSEs are now graded on a 1-9 scale.

p. 8 line 28. I wonder if there is a reliability/validity problem here? I would avoid a word like 'somewhat' with young children. Do they ever use such a word or even know what it means? I

could imagine some children guessing that ‘somewhat agree’ might mean ‘strongly agree’, which would mess up your scale.

p. 8 line 44. In a similar way, I am not even sure that I am clear myself about the difference between ‘partly true’ and ‘somewhat true’, so this seems like a particularly problematic choice of words for a scale. It is surprising the alpha is so high if people interpreted these differently? I suppose their presentation in order along a line meant that the words didn’t matter too much.

p. 9 line 23 – these names seem much better to me

p. 8 (c) (i) I think I need to know what the other statements were, as only a few examples are given. Otherwise, saying that ‘they’ reduced to 2 factors is a bit meaningless, as I don’t know what ‘they’ were – what variables reduced to 2 factors? Maybe all your statements just happened to be clearly about those 2 things? Could the complete lists be included in the appendices? Similarly on Line 53 you say that ‘Three factors were included’, but I think this language is a bit confusing, as this makes me think of factors emerging from a factor analysis, but these were measured using separate instruments.

p. 10 (e) (iv) Is there any reliability/validity data for this scale?

p. 18 “more positive attitudes equated to lower maths attainment” – I agree that this is very surprising. I suppose there is no possibility of a coding error or something?

bottom of p. 20 – top of p. 21 I don’t think it is so surprising that the stronger mathematicians may feel less at home at school and are to some extent social ‘misfits’. Regarding the subsequent discussion on p. 21, there may be findings in the PISA data to support strong maths students not liking the subject. Could it also be that stronger maths students are harsher critics of their teachers – the better you are at maths the more likely you may be to think that your teacher is a fool? 8-)

p. 22 top, maybe the stronger maths students are more likely to be picked on by other students and have a less happy experience in school?

I am just suggesting these things as they occur to me in case they help your discussion.

Small suggestions:

Consider ‘mathematics’ rather than ‘maths’ throughout

Would ‘primary’ and ‘secondary’ be simpler language for the reader than ‘pre-transition’ and ‘post-transition’?

p. 7, 2(b)(i) – not sure why there is (i), as there is no (ii)?

p. 13 Table 1 (and all the other tables) – the header row was shaded so darkly that I could hardly read the column headings (only by zooming in enormously)

p. 13 line 46 – I prefer to avoid using words like ‘very’ (especially in italics) in scientific writing – can the value be stated instead? Similarly, on p. 20 line 18 I was unsure how to interpret “somewhat substantial”???

p. 16 line 23 missing apostrophe in ‘years’

A couple of times you use the word “Although” where I think it is grammatically incorrect, and should be “However” (p. 16 Line 40; p. 18 line 38)

p. 17 Could the significant predictors in these tables be indicated in some way for the convenience of the reader?

Reviewer: 2

Comments to the Author(s)

This is an interesting manuscript with a useful and timely focus. It is very clearly structured, and is written clearly, concisely and in an engaging way. However, there are three issues that mean it

is not possible to recommend publication of the manuscript in its current form. These issues are to do with the framing of the study, the definition and justification of the variables included in the analysis, and with the interpretation of some of the findings. It may be that these issues are relatively straightforward to deal with – but they seem substantial enough to recommend major, rather than minor, revisions, especially as they relate to aspects across most sections of the current manuscript.

With regard to the framing of the research study, it is not clear to what extent the article is really about transition. Much of the introduction suggests this is a key focus – including Sections 1a and 1c. However, the discussion makes clear that actually the data available do not allow the authors to say much about the point of transition – much of the data is from when children were 14 years old and had already been at their secondary school for 2-3 years. Also, Stage-Environment Fit theory is not mentioned again after the introduction, which suggests that it was not a useful tool to interpret the results of the analysis. I recommend that the emphasis on the point of transition be reduced in the introduction, and that a more consistent focus is placed on relationships between attitudes and attainment throughout the paper – including on whether we might expect relationships between attitudes and attainment to be similar or different in secondary school compared to primary school.

With regard to the measures, there is a need for some more information in some areas and some justification for the inclusion of some measures in others. More information is certainly needed about the mathematics achievement measures used. It is not possible to tell from the text in Section 2bi what variables have actually been used from the NPD. Readers will therefore not be able to know what the data represent – whether they are national standardised tests or teacher assessments, or some combination of the two. Actual variable names from the NPD should be given – e.g. for KS3 outcome – was this NC level, or paper test score(s), or teacher assessment, or a combination of these? I would think that total score on tests would be the most valid measure – i.e. KS3.KS3_MATTOTMRK – as it is clear what this represents and this variable is not subject to teacher bias in the same way as a teacher assessment of NC level may be.

A further issue with the measures is a need for clearer articulation between the literature review and the variables included in the analysis. In most cases this is clear, but there are a few where the connection is not clear. The main examples of this are the teacher characteristics included in the pre-transition model. On page 6, the manuscript states, “In addition to the effects associated with a positive student-teacher relationship, teachers’ own attitudes, self-efficacy beliefs, and abilities can influence the development of students’ attitudes towards maths regarding gender stereotypes [57], and can affect students’ attainment [57,58]. Teachers’ enjoyment of maths also affects the instructional time given to maths in that teachers who enjoy maths more spend more time engaging in maths tasks [59].” However, none of these factors seem to connect directly with teachers’ own mental health as measured with the CCEI or BSE measures. Therefore it seems that these could have just been included because they are available in the dataset – some clearer justification for their inclusion should be given.

Finally, with regard to the interpretation of some of the findings in the discussion section, there is a need for some more detailed analysis – particularly of the more unexpected findings. In the current discussion, it is not sufficiently clear to what extent the authors think that these are informative or important to research further, given the data drawn on and the literature that hypotheses were initially based on. For example, there is a difference in the effect of relationship with teachers on attainment, between 11 years old and 14 years old. When this is first discussed, in Section 4a on page 20, the text would benefit from a more thorough analysis of potential reasons why this may be. These reasons would include the possibility that it could be a genuine effect, but also that it could be due to the different measures included in the two models, or differences in structural aspects of primary versus secondary education in England (including the fact that students generally have 1 teacher in primary school, but many teachers in secondary school). Again, clearer articulation between sections in the manuscript would help the reader in making sense of the findings – for example in the introduction, primary school teachers’ attitudes

to mathematics are mentioned as an important factor, but do not appear to have been included in the pre-transition model (presumably because this data is not available). This is fine, but it would help the reader if this was discussed more explicitly in the manuscript.

Some further development of the text in response to these issues would make for a much stronger contribution to the literature. The core concepts of the work are sound. In its current form, the value of the research is just not as clear as it could be.

Author's Response to Decision Letter for (RSOS-200975.R0)

See Appendix A.

RSOS-200975.R1 (Revision)

Review form: Reviewer 1

Is the manuscript scientifically sound in its present form?

Yes

Are the interpretations and conclusions justified by the results?

Yes

Is the language acceptable?

Yes

Do you have any ethical concerns with this paper?

No

Have you any concerns about statistical analyses in this paper?

No

Recommendation?

Accept as is

Comments to the Author(s)

The authors have addressed my previous concerns and I recommend this paper for publication.

Small remaining suggestions:

p. 13 line 8, delete 'in'

p. 14 line 43, 'post-transition' could be 'secondary'

p. 14 lines 48-50, again the 'pre-transition' and 'post-transition' terminology could be primary/secondary here as well, as changed elsewhere

p. 16 line 36 'reports' should be 'report'

p. 16 line 38 it seems odd to define 'positive social context' with a bracket full of negative things

Review form: Reviewer 2

Is the manuscript scientifically sound in its present form?

Yes

Are the interpretations and conclusions justified by the results?

Yes

Is the language acceptable?

Yes

Do you have any ethical concerns with this paper?

No

Have you any concerns about statistical analyses in this paper?

No

Recommendation?

Accept as is

Comments to the Author(s)

The authors have done a very thorough job in responding to the issues that I raised in my previous review. The additional information and discussion is very helpful.

As before, I think this is a very well-written and informative paper, that makes a valuable contribution to the field. I am very happy to recommend publication at this stage.

Decision letter (RSOS-200975.R1)

Dear Ms Evans,

It is a pleasure to accept your manuscript entitled "Maths attitudes, school affect, and teacher characteristics as predictors of maths attainment trajectories in primary and secondary education" in its current form for publication in Royal Society Open Science. The comments of the reviewer(s) who reviewed your manuscript are included at the foot of this letter.

Best regards,

on behalf of Dr Emma Hayiou-Thomas (Associate Editor) and Essi Viding (Subject Editor)
openscience@royalsociety.org

Associate Editor Comments to Author (Dr Emma Hayiou-Thomas):

Thank you for your careful responses to the previous round of reviews, which have further strengthened the ms. This is a well-executed and very clearly written study, which I'm confident will be of interest to the field. I'm happy to recommend it for publication at this stage - please note some very minor suggestions re wording in Reviewer 1's comments.

Reviewer comments to Author:

Reviewer: 1
Comments to the Author(s)

The authors have addressed my previous concerns and I recommend this paper for publication.

Small remaining suggestions:

- p. 13 line 8, delete 'in'
- p. 14 line 43, 'post-transition' could be 'secondary'
- p. 14 lines 48-50, again the 'pre-transition' and 'post-transition' terminology could be primary/secondary here as well, as changed elsewhere
- p. 16 line 36 'reports' should be 'report'
- p. 16 line 38 it seems odd to define 'positive social context' with a bracket full of negative things

Reviewer: 2
Comments to the Author(s)

The authors have done a very thorough job in responding to the issues that I raised in my previous review. The additional information and discussion is very helpful. As before, I think this is a very well-written and informative paper, that makes a valuable contribution to the field. I am very happy to recommend publication at this stage.

Appendix A

Author response to reviews of

Manuscript RSOS-200975

Maths attitudes, school affect, and teacher characteristics as predictors of maths attainment trajectories in primary and secondary education

submitted to *Royal Society Open Science*

RC: Reviewer Comment AR: Author Response Manuscript text

Dear Dr Hayiou-Thomas,

Thank you very much for taking the time to consider our manuscript for publication at *Royal Society Open Science*. In the following we address the reviewers' concerns and suggestions, and describe the revisions made to the manuscript.

1. Reviewer #1

RC: Intro – This is framed as a problem of maths attainment, and yet the statistics about few adults having skills commensurate with those of children suggests that the problem may be as much about retention over time as about the initial learning in school?

AR: The below sentence has been added to the introduction on page 10.

The poor maths performance seen in adults in the UK is likely due to deficits in childhood learning but could also be due to poor retention or a lack of practice of maths skills over time (see Geary, 2000).

RC: It is repeatedly stated (e.g. p. 3; p. 5 line 50) that “Children have several specialised subject teachers in secondary education compared to one individual teacher for all subjects for the entire school-year in primary education.” This is less true now in England than it was, especially in maths, where specialist maths teachers are increasingly common in primary, so this may need to be nuanced.

AR: Thank you for highlighting this, the two sentences have been changed to the following:

Children generally have several specialised subject teachers in secondary education compared to one individual teacher for all subjects for the entire school-year in primary education (though the presence of specialist maths teachers is becoming increasingly common in English primary schools, helped by government initiatives and training bursaries). (pg 12)

In primary education, children are traditionally taught by a single teacher per year for all subjects (though primary schools are increasingly utilising specialist teachers in recent years) whereas... (pg 17)

RC: p. 4. There is perhaps a bit of slippage into ‘causal’ language here, with ‘negative effects’ and talk about interventions, and ‘beneficial’ (line 38), which is all carefully avoided in most places elsewhere.

AR: The first line has been removed, and ‘an academically-focused environment is beneficial for children’s general academic and maths attainment’ changed to ‘an academically-focused environment is positively associated with children’s general academic and maths attainment’.

RC: p. 4 line 49 “much greater teacher to student ratio” – I think ‘lower’ is what is meant here? But I don’t think this is true. Certainly not ‘much’. Classes are capped at 30 in primary, but are rarely larger than this on average at secondary, in my experience.

AR: The following ‘The number of students also increases significantly from primary to secondary education, with a much greater teacher-to-student ratio, meaning that children have a decreased capacity to develop similar relationships and attachments that they had with their teachers in primary education (Eccles et al., 1993).’ has been changed to:

The total number of students also increases significantly from primary to secondary education, with teachers typically interacting with multiple classes of children in different years throughout the school day, meaning that children have a decreased capacity to develop close relationships and attachments like they had with their teachers in primary education (Eccles et al., 1993). (pg 14-15)

RC: p. 7, 2(a)(i) – I was surprised that SEND children were excluded. Were there not enough of these to do a separate subgroup analysis? You might find larger effects for this group, perhaps? I think schools would be interested in the findings of this.

AR: We did not conduct separate subgroup analyses for SEN children because missing data was higher in this group and we also didn’t have access to the individual SEN status of each child (i.e. their specific diagnosis or difficulty), and because of the high heterogeneity within this group where some children would have ‘typical’ maths attainment and abilities and others may have significant difficulties with maths, we felt it would be inaccurate to assess this group without knowledge of their individual difficulties.

RC: p. 7 lines 54-55 these ages are for the END of key stage years (e.g., Year 9 for the end of KS3, but KS3 is years 7-9, so ages 11-14).

AR: This has been changed from ‘The maths attainment of primary- and secondary-education students in England is measured through examinations and assessments at the end of each ‘Key Stage’. There are 4 key stages throughout children’s compulsory education, with key stage 1 (age 6-7) and key stage 2 (age 10-11) in primary education, and key stage 3 (age 13-14) and key stage 4 (age 15-16) in secondary education.’ to:

There are four key stages throughout children's compulsory education in England, with key stage 1 (age 5-7) and key stage 2 (age 7-11) in primary education, and key stage 3 (age 11-14) and key stage 4 (age 14-16) in secondary education. The maths attainment of primary- and secondary-education students is measured through examinations and assessments at the end of each key stage (i.e. at age 6-7, 10-11, 13-14, and 15-16). (pg 22)

RC: p. 7-8, 2(b)(i) **Some of this information is out of date. There are no longer KS3 tests. And GCSEs are now graded on a 1-9 scale.**

AR: The data were collected when KS3 tests still existed and before the introduction of the newer GCSE grading criteria, but this has been noted in the method section as per the below:

It is important to highlight that this scoring differs to the current grading system in England where key stage 3 tests are no longer administered, and where key stage 4 assessments are graded on a 1-9 scale. (pg 23)

RC: p. 8 line 28. **I wonder if there is a reliability/validity problem here? I would avoid a word like 'somewhat' with young children. Do they ever use such a word or even know what it means? I could imagine some children guessing that 'somewhat agree' might mean 'strongly agree', which would mess up your scale.**

p. 8 line 44. In a similar way, I am not even sure that I am clear myself about the difference between 'partly true' and 'somewhat true', so this seems like a particularly problematic choice of words for a scale. It is surprising the alpha is so high if people interpreted these differently? I suppose their presentation in order along a line meant that the words didn't matter too much.

AR: These are both very valid points. The scales were created by ALSPAC and so we had no control over the design of the questions/responses. For all of the questions, the responses were presented as boxes to tick where the responses were ordered, i.e. 'not true' to 'true' from left to right, so it should have been *relatively* easy for children to infer the meaning of the statements from the way they were presented (if they were unsure of the difference between 'partly true' and 'somewhat true' for example).

RC: p. 8 (c) (i) **I think I need to know what the other statements were, as only a few examples are given. Otherwise, saying that 'they' reduced to 2 factors is a bit meaningless, as I don't know what 'they' were – what variables reduced to 2 factors? Maybe all your statements just happened to be clearly about those 2 things? Could the complete lists be included in the appendices? Similarly on Line 53 you say that 'Three factors were included', but I think this language is a bit confusing, as this makes me think of factors emerging from a factor analysis, but these were measured using separate instruments.**

AR: A list of the individual items will be made available in supplementary material (indicated in the text in the first paragraph of the section 'Substantial predictors: primary education').

The sentence 'Factors related to teacher characteristics were assessed in the final year of primary education (in year 6; when children are age 10-11). Three factors were included, . . .' has been changed to:

Measures related to teacher characteristics were assessed in the final year of primary education (in year 6; when children are age 10-11). Three variables were included, . . . (pg 24)

RC: p. 10 (e) (iv) Is there any reliability/validity data for this scale?

AR: Added in:

The AMCIES has shown good reliability in other samples (Cronbach's $\alpha = .76-.80$; Jaekel, Wolke & Chernova, 2012). (pg 29)

RC: p. 18 “more positive attitudes equated to lower maths attainment” – I agree that this is very surprising. I suppose there is no possibility of a coding error or something?

AR: There was a coding error which resulted in the association being reversed and has now been rectified. The results and discussion have been updated accordingly. Thank you for spotting this!

RC: bottom of p. 20 – top of p. 21 I don't think it is so surprising that the stronger mathematicians may feel less at home at school and are to some extent social 'misfits'. Regarding the subsequent discussion on p. 21, there may be findings in the PISA data to support strong maths students not liking the subject. Could it also be that stronger maths students are harsher critics of their teachers – the better you are at maths the more likely you may be to think that your teacher is a fool? 8-)

p. 22 top, maybe the stronger maths students are more likely to be picked on by other students and have a less happy experience in school?

AR: Thank you for these suggestions! The below has been updated to include them.

From 'However, it is particularly surprising that student-reported school-belonging was negatively associated with attainment. It appears that the school-climate declines around the transition (Coelho et al., 2020), however, this still does not explain the negative association with maths as found here. One possible explanation is that the measure used for school belonging in this study contained items relating to the child's peer relationships, and as such, could reflect their perceived popularity. For example, items for the school belonging composite included 'my school is a place where I know people who think a lot of me', 'my school is a place where I get on well with other pupils in my classes', and 'my school is a place where other pupils are very friendly'. Therefore, it could be that adolescents who perceive their peers as more accepting and friendly, are those with a greater number of friendships and are considered 'popular', which has been associated with decreased attainment (Schwartz et al., 2006). However, further research is needed to investigate this possibility.' to

However, it is somewhat surprising that student-reported school-belonging was negatively associated with attainment. It appears that the school-climate declines around the transition (Coelho et al., 2020), however, this still does not explain the negative association with maths as found here. One possible explanation is that the measure used for school belonging in this study contained items relating to the child's peer relationships, and as such, could reflect their perceived popularity. For example, items for the school belonging composite included 'my school is a place where I know people who think a lot of me', 'my school is a place where I get on well with other pupils in my classes', and 'my school is a place where other pupils are very friendly'. Therefore, it could be that adolescents who perceive their peers as more accepting and friendly, are those with a greater number of friendships and are considered 'popular', which has been associated with decreased attainment (Schwartz, Gorman, Nakamoto, & McKay, 2006). Another potential explanation could be that students who are especially 'gifted' in maths may not feel comfortable socially within their school or may not find it sufficiently challenging intellectually. Peer victimisation is high for gifted students (Peterson & Ray, 2006), and so it could be that high-achieving maths students do not view their school as a place they get on well with other pupils. In addition, stronger mathematicians may feel less engaged by classwork they do not find particularly challenging, and so may not identify strongly with their school and their educational environment. However, additional research is needed to investigate these possibilities further. (pg 40-41)

RC: Consider 'mathematics' rather than 'maths' throughout.

AR: This study is the final part of a three-phase investigation, and because the other two papers have both used 'maths' throughout, we will keep it as 'maths' here too for consistency between the papers.

RC: Would 'primary' and 'secondary' be simpler language for the reader than 'pre-transition' and 'post-transition'?

AR: This has been changed throughout.

RC: p. 7, 2(b)(i) – not sure why there is (i), as there is no (ii)?

p. 13 Table 1 (and all the other tables) – the header row was shaded so darkly that I could hardly read the column headings (only by zooming in enormously)

AR: Apologies, both of these issues are to do with the LaTeX class used.

RC: p. 13 line 46 – I prefer to avoid using words like 'very' (especially in italics) in scientific writing – can the value be stated instead? Similarly, on p. 20 line 18 I was unsure how to interpret "somewhat substantial"???

AR: First line has been removed, and the second has been changed to state 'moderate' - the size of the effect is given in the following line.

RC: p. 16 line 23 missing apostrophe in 'years'

AR: Thanks for noticing! This has been corrected.

RC: A couple of times you use the word "Although" where I think it is grammatically incorrect, and should be "However" (p. 16 Line 40; p. 18 line 38)

AR: Both have been changed to 'however'.

RC: p. 17 Could the significant predictors in these tables be indicated in some way for the convenience of the reader?

AR: Because so many of the correlations are significant due to the very large sample size, we felt it was misleading to use marked significance in the tables where the actual effect size is very small.

2. Reviewer #2

RC: With regard to the framing of the research study, it is not clear to what extent the article is really about transition. Much of the introduction suggests this is a key focus – including Sections 1a and 1c. However, the discussion makes clear that actually the data available do not allow the authors to say much about the point of transition – much of the data is from when children were 14 years old and had already been at their secondary school for 2-3 years. Also, Stage-Environment Fit theory is not mentioned again after the introduction, which suggests that it was not a useful tool to interpret the results of the analysis. I recommend that the emphasis on the point of transition be reduced in the introduction, and that a more consistent focus is placed on relationships between attitudes and attainment throughout the paper – including on whether we might expect relationships between attitudes and attainment to be similar or different in secondary school compared to primary school.

AR: The focus on the transition in the introduction (predominantly in section 1a and 1c) has been reduced and reframed to highlight the transition as a period of change and discusses the differences between primary and secondary education with less emphasis on the transition event as the key focus of the study. Greater emphasis has been placed upon why we might expect the relationship between the predictors and maths to differ between primary and secondary education in these sections also (pg 11-14). The recommendation of reviewer 1 to use ‘primary’ and ‘secondary’ throughout rather than ‘pre-transition’ and ‘post-transition’ has also been incorporated which should reduce the focus on the transition given that the measures do not correspond directly with this time period.

Correspondingly, the abstract has been altered (changed from ‘pre’- and ‘post-transition’, to ‘primary’ and ‘secondary education’), and the title has also been amended to be ‘Maths attitudes, school affect, and teacher characteristics as predictors of maths attainment trajectories in primary and secondary education’.

The stage-environment fit theory is now included in the discussion as a potential explanation for why significant associations were found for variables in secondary education but not in primary education (pg 43).

RC: With regard to the measures, there is a need for some more information in some areas and some justification for the inclusion of some measures in others. More information is certainly needed about the mathematics achievement measures used. It is not possible to tell from the text in Section 2bi what variables have actually been used from the NPD. Readers will therefore not be able to know what the data represent – whether they are national standardised tests or teacher assessments, or some combination of the two. Actual variable names from the NPD should be given – e.g. for KS3 outcome – was this NC level, or paper test score(s), or teacher assessment, or a combination of these? I would think that total score on tests would be the most valid measure – i.e. KS3.KS3_MATTOTMRK – as it is clear what this represents and this variable is not subject to teacher bias in the same way as a teacher assessment of NC level may be.

AR: The measures for maths were national curriculum levels obtained by ALSPAC from the NPD which is now explicitly stated in the method section along with the NPD variable names:

In key stages 1-3, children's progress is evaluated using national curriculum levels which are a numerical grades ranging from 1-8, with a higher score indicative of greater maths attainment. Governmental guidelines suggest that it is expected that children achieve a level 2 at key stage 1, a level 4 at key stage 2, and between levels 5-6 at key stage 3. At key stage 4, adolescents can achieve an alphabetical grade from the highest of 'A*', through 'A', 'B', 'C', 'D', 'E', 'F', 'G', and the lowest grade of a 'U'. To be comparable to maths attainment at the other key stages, these alphabetical grades were coded into numerical grades with the highest being grade 10 (i.e. 'A*'), down to the lowest grade of 2 (i.e. 'U'). National curriculum levels for maths were obtained by ALSPAC from local education authorities for key stage 1 data, and the NPD for key stage 2-4 data (NPD variables: K2_LEVM, K3_LEVM, and KS4_APMAT), which consisted of a combination of teacher assessments and standardised tasks and tests. It is important to highlight that this scoring differs to the current grading system in England where key stage 3 tests are no longer administered, and where key stage 4 assessments are graded on a 1-9 scale. (pg 22-23)

RC: A further issue with the measures is a need for clearer articulation between the literature review and the variables included in the analysis. In most cases this is clear, but there are a few where the connection is not clear. The main examples of this are the teacher characteristics included in the pre-transition model. On page 6, the manuscript states, "In addition to the effects associated with a positive student-teacher relationship, teachers' own attitudes, self-efficacy beliefs, and abilities can influence the development of students' attitudes towards maths regarding gender stereotypes [57], and can affect students' attainment [57,58]. Teachers' enjoyment of maths also affects the instructional time given to maths in that teachers who enjoy maths more spend more time engaging in maths tasks [59]." However, none of these factors seem to connect directly with teachers' own mental health as measured with the CCEI or BSE measures. Therefore it seems that these could have just been included because they are available in the dataset – some clearer justification for their inclusion should be given.

AR: This has been rewritten as the following:

In addition to the effects associated with a positive student-teacher relationship, teachers' own attitudes, self-efficacy beliefs, and abilities can influence the development of students' attitudes towards maths regarding gender stereotypes (Gunderson, Ramirez, Levine, & Beilock, 2012), and can affect students' attainment (Gunderson et al., 2012; Thomson, Walkowiak, Whitehead, & Huggins, 2020). Teachers' enjoyment of maths also affects the instructional time given to maths in that teachers who enjoy maths more spend more time engaging in maths tasks (Russo et al., 2020). There is also some evidence to suggest teachers' general mental wellbeing is linked to students' maths abilities through the quality of the classroom learning environment (McLean & Connor, 2015), and in the feedback given to students (McLean & Connor, 2018), which is particularly marked for low-achieving students. Research in this area is sparse, but generally suggests that teachers' mental health and their attitudes towards maths are linked to students' maths outcomes. (pg 17)

RC: Finally, with regard to the interpretation of some of the findings in the discussion section, there is a need for some more detailed analysis – particularly of the more unexpected findings. In the current discussion, it is not sufficiently clear to what extent the authors think that these are informative or important to research further, given the data drawn on and the literature that hypotheses were initially based on. For example, there is a difference in the effect of relationship with teachers on attainment, between 11 years old and 14 years old. When this is first discussed, in Section 4a on page 20, the

text would benefit from a more thorough analysis of potential reasons why this may be. These reasons would include the possibility that it could be a genuine effect, but also that it could be due to the different measures included in the two models, or differences in structural aspects of primary versus secondary education in England (including the fact that students generally have 1 teacher in primary school, but many teachers in secondary school). Again, clearer articulation between sections in the manuscript would help the reader in making sense of the findings – for example in the introduction, primary school teachers’ attitudes to mathematics are mentioned as an important factor, but do not appear to have been included in the pre-transition model (presumably because this data is not available). This is fine, but it would help the reader if this was discussed more explicitly in the manuscript.

AR: The discussion section now includes a discussion of the implications of the findings on page 45-46, where we highlight the application of the key findings and the limitations of the analyses (i.e. not being able to establish causality) that affect the interpretation of the results. Greater discussion on the differing associations between variables in the primary and secondary education models has been added on page 43.

Many measures that would have been of interest were not available in the ALSPAC dataset, which is now stated more clearly within the discussion section:

The application and interpretation of these findings are affected by methodological limitations that warrant further discussion. Firstly, this study aimed to focus on the effects of the transition to secondary education, comparing school-related factors across the transition to secondary education and how they may affect maths attainment trajectories. However, the timings of the measures used are not directly before and following transition, meaning that we cannot say with any certainty that the transition event itself had a direct impact on the outcomes. Additionally, the measures are not directly comparable pre- and post-transition meaning that we were unable to look at any changes over time in school-related affect, student-teacher relationships, and maths attitudes. Another weakness of the study was due to the availability of measures within the ALSPAC dataset. It was not possible to include several measures that have been linked to maths outcomes in previous studies (such as teachers’ attitudes towards maths and their maths anxiety for example), or to include variables that were measured in both primary or secondary education (i.e. attitudes towards maths teacher). Including these predictors would have been useful in obtaining a more comprehensive model of maths attainment trajectories, and by assessing the relative importance of different factors for attainment in primary and secondary education. (pg 46)

We are extremely appreciative of your reviews especially under the current circumstances and hope that these amendments are responsive to your recommendations.

Yours sincerely

Danielle Evans